# A Review of the Optimal Design of Neural Networks Based on FPGA

**Chenghao Wang** [1] **and Zhongqiang Luo** [1,2,*]

1 School of Automation and Information Engineering, Sichuan University of Science and Engineering, Yibin 644000, China
2 Artificial Intelligence Key Laboratory of Sichuan Province, Sichuan University of Science and Engineering, Yibin 644000, China
* Correspondence: luozhongqiang@suse.edu.cn

**Abstract:** Deep learning based on neural networks has been widely used in image recognition, speech recognition, natural language processing, automatic driving, and other fields and has made breakthrough progress. FPGA stands out in the field of accelerated deep learning with its advantages such as flexible architecture and logic units, high energy efficiency ratio, strong compatibility, and low delay. In order to track the latest research results of neural network optimization technology based on FPGA in time and to keep abreast of current research hotspots and application fields, the related technologies and research contents are reviewed. This paper introduces the development history and application fields of some representative neural networks and points out the importance of studying deep learning technology, as well as the reasons and advantages of using FPGA to accelerate deep learning. Several common neural network models are introduced. Moreover, this paper reviews the current mainstream FPGA-based neural network acceleration technology, method, accelerator, and acceleration framework design and the latest research status, pointing out the current FPGA-based neural network application facing difficulties and the corresponding solutions, as well as prospecting the future research directions. We hope that this work can provide insightful research ideas for the researchers engaged in the field of neural network acceleration based on FPGA.

**Keywords:** deep learning; deep neural network; FPGA; optimization; hardware acceleration

## 1. Introduction

With the rise of the short video industry and the advent of the era of big data and the Internet of Things (IoTs), the data created by people in recent years has shown a blowout growth, providing a solid data foundation for the development of artificial intelligence (AI). As the core technology and research direction for realizing artificial intelligence, deep learning based on neural networks has achieved good results in many fields, such as speech recognition [1–3], image processing [4–6], and natural language processing [7–9]. Common platforms to accelerate deep learning include central processing unit (CPU), graphics processing unit (GPU), field-programmable gate array (FPGA), and application-specific integrated circuit (ASIC).

Among them, the CPU adopts Von Neumann architecture and the program execution in the field of deep learning of artificial intelligence is less, while the computational demand for data is relatively large. Therefore, the implementation of AI algorithms by CPU has a natural structural limitation—that is, the CPU spends a large amount of time reading and analyzing data or instructions. In general, it is not possible to achieve unlimited improvement of instruction execution speed by increasing CPU frequency and memory bandwidth without limit.

The ASIC special purpose chip has the advantages of high throughput, low latency, and low power consumption. Compared with FPGA implementation of the same process, ASIC can achieve 5~10 times the computation acceleration, and the cost of ASIC will be greatly reduced after mass production. But, deep learning computing tasks are flexible, each algorithm's needs can be implemented effectively through slightly different dedicated hardware architecture, and there is a high cost of research and development of ASIC and flow, with the cycle being long, yet the most important thing to note is that is logic cannot cause dynamic adjustment, which means that the custom accelerator chips (such as ASIC) must make a large amount of compromise in order to act as a sort of universal accelerator.

At a high level, this means that system designers face two choices. One is to select a heterogeneous system and load a large number of ASIC accelerator chips into the system to deal with various types of problems. Or, one can choose a single chip to handle as many types of algorithms as possible. The FPGA scheme falls into the latter category because FPGA provides unlimited reconfigurable logic, whereas FPGA can update the logic function in only a few hundred milliseconds, and we can design accelerators for each algorithm. The only compromise is that the accelerators involve programmable logic rather than a hardened gate, but this also means that we can take advantage of the flexibility of FPGA to help us in saving development costs even more.

At present, artificial intelligence computing requirements represented by deep learning mainly use GPU, FPGA, and other existing general chips that are suitable for parallel computing in order to achieve acceleration. Due to the characteristics of high parallelism, high frequency, and high bandwidth, GPU can parallelize the operation and greatly shorten the operation time of the model. Due to its powerful computing ability, it is mainly used to deal with large-scale computing tasks at present. GPU-accelerated deep learning algorithms have been widely used and have achieved remarkable results.

Compared with FPGA, the peak performance of GPU (10 TFLOPS, floating point operation per second) is much higher than that of FPGA (<1 TFLOPS), and thus GPU is a good choice for deep learning algorithms in terms of accelerating performance. However, this is only the case when the power consumption is not considered. Because the power consumption of GPU is very high, sometimes tens of hundreds of times that of CPU and FPGA, the high energy consumption limits it to being used in high-performance computing clusters. If it is used on edge devices, the performance of GPU will be greatly sacrificed.

However, the DDR (double data rate) bandwidth, frequency, and number of computing units (low-end chips) of FPGA are not as high as GPU. For on-chip memory, FPGA has larger computing capacity; moreover, on-chip memory is crucial for reducing latency in applications such as deep learning. Accessing external memory such as DDR consumes more energy than the chip itself for computing. Therefore, the larger capacity of on-chip cache reduces the memory bottleneck caused by external memory reading and also reduces the power consumption and cost required by high-memory bandwidth. The large capacity of on-chip memory and flexible configuration capability of FPGA reduces the read and write of external DDR, while GPU requires the support of an external processor during operation. The addition of external hardware resources greatly reduces the data processing speed. Moreover, FPGA's powerful raw data computing power and reconfigurability allow it to process arbitrary precision data, but GPU's data processing is limited by the development platform. Therefore, in this context, FPGA seems to be a very ideal choice. In addition, compared with GPU, FPGA not only has the characteristics of data parallel, but also has the characteristics of pipeline parallel, and thus for pipelined computing tasks, FPGA has a natural advantage in delay. On the basis of these advantages, FPGA stands out in the field of accelerated deep learning.

In the previous work, the application, acceleration method and accelerator design of neural networks such as CNN (convolutional neural network), RNN (recurrent neural network), and GAN (generative adversarial network) based on FPGA were described in

detail, and the research hotspots of industrial application combined with FPGA and deep neural networks were fully investigated. They have made relevant contributions to the application and development of FPGA in neural networks [10–14]. It is worth noting that the analysis of the acceleration effect of different acceleration techniques on different neural networks is less involved in their work. Motivated by the current research progress, this paper reviews the latest optimization technology and application schemes of various neural networks based on FPGA, compares the performance effect of different acceleration technologies on different neural networks, and analyzes and summarizes of potential research prospects.

Compared with the previous work, the contributions of this paper are as follows:

(1) The development history of neural networks is divided into five stages and presented in the form of tables, so that readers can understand the development history of neural networks more intuitively.

(2) Study of the optimization technology of various neural networks based on FPGA, and introducing the application scenarios of various technologies and the latest research results of various technologies.

(3) Introducing the latest application achievements of CNN, RNN, and GAN in the field of FPGA acceleration, analyzing the performance achieved by deploying different neural networks using different optimization technologies on different FPGA platforms, and finding that the use of Winograd and other convolutional optimization technologies can bring about huge performance gains. The reasons for this phenomenon are analyzed.

(4) The future research directions of neural network acceleration based on FPGA are pointed out, and the application of FPGA accelerated neural network is prospected.

The remainder of this paper is organized as follows: In Section 2, the history of neural networks is clearly shown in the form of a table. The history of neural networks is divided into five stages, being convenient for readers to comb and study. In Section 3, some common neural networks are introduced, and their applications based on FPGA and their effects are described. In Section 4, the acceleration technology of various neural networks based on FPGA is introduced, its advantages and disadvantages and application scenarios are pointed out, and the latest research status and achievements are described. In Section 5, this paper describes the FPGA-based neural network accelerator and the acceleration framework in detail. These accelerators are often the comprehensive application of the acceleration technology described in Section 4, and they are compared with and summarized in terms of the performance of different neural networks deployed on different FPGA platforms using different acceleration technologies. In Section 6, the current difficulties in the application of FPGA-based neural networks are pointed out, and the future research directions are prospected. Finally, the paper is summarized in Section 7. The structural layout of the article is illustrated in Figure 1.

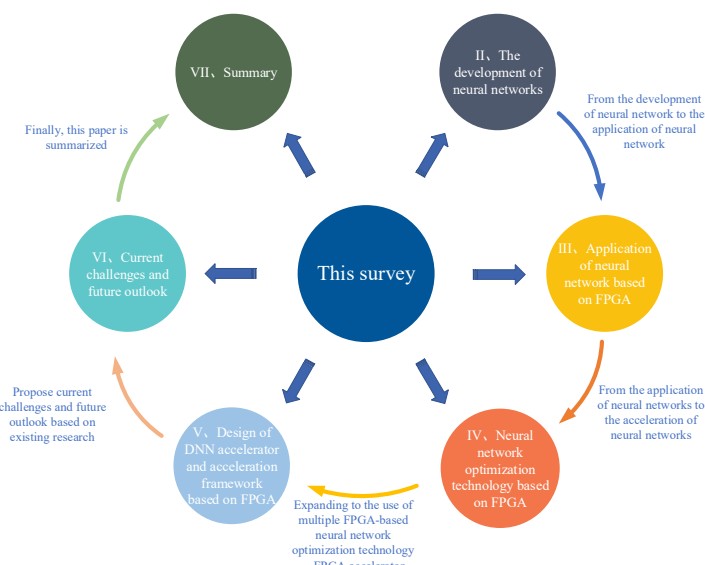

**Figure 1.** The structural layout of this paper.

## 2. The Development of Neural Networks

As shown in Table 1, the development process of neural networks can be roughly divided into five stages: the proposal of the model, the stagnation period, the rise of the back propagation algorithm, the confusion period, and the rise of deep learning.

The period of 1943–1958 is considered as the model proposal phase, and the perceptron wave proposed in 1958 continued for about 10 years. With increasingly more scholars entering into this direction of research, some scholars gradually found the limitations of perceptron model. With the publication of the Perceptron by M. Minsky in 1969, the neural network was pushed to the bottom directly, leading to the "stagnation period" of neural networks from 1969 to 1980.

After 1980, increasingly more scholars paid attention to the back propagation algorithm, which reopened the thinking for researchers and launched another spring for the development of neural networks.

Then, for a long period of time, scholars did not make breakthrough results, only working on the basis of existing research. In the mid-1990s, statistical learning theory and the machine learning model represented by the support vector machine began to rise. In contrast, the theoretical basis of neural networks was not clear, and optimization difficulties, poor interpretability, and other shortcomings became more prominent; thus, neural network research fell into a low tide again.

Until 2006, Professor Geoffrey Hinton, a neural network expert at the University of Toronto, and his students formally proposed the concept of deep learning. They proposed the deep belief network model in their published paper. Hinton et al. found that the multi-layer feedforward neural network could be pre-trained layer by layer. In other words, the unsupervised pre-training method is used to improve the initial value of the network weights, and then the weights are fine-tuned. This model began the research boom of deep neural networks and opened the prelude to the research and application of deep learning.



**Table 1.** Development history of neural networks.

| Stage | Year | Character | Content |
|---|---|---|---|
| Generation of models | 1943 | Warren McCulloch and Walter Pitts | McCulloch–Pitts model [15] |
| | 1948 | Alan Mathison Turing | B-type Turing machine [16] |
| | 1949 | Donald Hebb | Hebb algorithm [17] |
| | 1951 | McCulloch and Marvin Minsky | The first neural network machine SNARC |
| | 1958 | Frank Rosenblatt | Perceptron model [18] |
| Lag phase | 1969 | Marvin Minsky | Perception [19] |
| | 1974 | Paul J. Werbos | Backpropagation (BP) algorithm [20] |
| | 1980 | Kunihiko Fukushima | Neocognitron model [21] |
| The rise of backpropagation algorithms | 1982 | John J. Hopfield | Hopfield model [22] |
| | 1985 | Hinton and Sejnowski | Boltzmann machine [23] |
| | 1986 | David Rumelhart and James McClelland | Redescription of the BP algorithm [24] |
| | 1989 | LeCun | Introducing the BP algorithm to the convolutional neural network [25] |
| Confusion period | 1990–2005 | The rise of machine learning models has brought about great challenges to the development of neural networks | |
| The rise of deep learning | 2006 | Geoffrey Hinton | Deep Belief Networks [26] |
| | 2012 | Alex Krizhevsky | AlexNet [27] |
| | 2014 | Christian Szegedy | GoogLeNet [28] |
| | | Visual Geometry Group and Google DeepMind | VGGNet [29] |
| | | Ian J. Goodfellow | GAN [30] |
| | | Yi Sun and Xiaogang Wang | DeepID [31] |
| | | Ross Girshick and Jeff Donahue | Region-CNN (RCNN) [32] |
| | 2015 | Joseph Redmon | You Only Look Once (YOLOv1) [33] |
| | 2016 | AlphaGo, an artificial intelligence machine developed by Google's DeepMind, beat Go world champion Lee Sedol 4-1 | |
| | | Joseph Redmon | YOLOv2 [34] |
| | 2018 | Joseph Redmon | YOLOv3 [35] |
| | 2020 | Alexey Bochkovskiy | YOLOv4 [36] |
| | 2020–2022 | YOLOv5, YOLOv6 [37], YOLOv7 [38], etc. | |

At present, common deep learning models include deep neural networks (DNNs), convolutional neural networks (CNNs), recurrent neural networks (RNNs), and generative adversarial networks (GANs), among others.

*2.1. Deep Neural Network (DNN)*

Since a single-layer perceptron cannot solve linear inseparability problems, it cannot be used in industry. Scholars have expanded and improved the perceptron model by increasing the number of hidden layers and corresponding nodes to enhance the expression ability of the model, and thus the deep neural network (DNN) was created. The DNN is sometimes called a multi-layer perceptron (MLP). The emergence of the DNN overcomes

the low performance of a single-layer perceptron. According to the position of different layers in the DNN, the neural network layers inside the DNN can be divided into three types: the input layer, hidden layer, and output layer, as shown in Figure 2. Generally speaking, the first layer is the input layer, the last layer is the output layer, and the middle layers are all hidden layers. Layer to layer is fully connected, that is, any neuron in layer n must be connected with any neuron in layer n+1. Although the DNN appears complicated, from a small local model, it is the same as the perceptron, namely, a linear relation $z = \sum W_i X_i + b$ plus an activation function $\sigma(z)$.

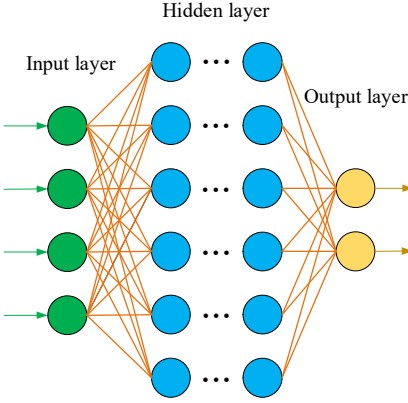

**Figure 2.** Typical model of a DNN.

With the deepening of the layers of neural networks, the phenomena of overfitting, gradient explosion, and gradient disappearance has become increasingly more serious, and the optimization function is increasingly more likely to fall into the local optimal solution. In order to overcome this problem, scholars propose convolutional neural networks (CNN) based on the receptive field mechanism in biology.

*2.2. Convolutional Neural Network (CNN)*

The CNN model was developed from the early artificial neural network. On the basis of the research of Hub et al. on the cells in the visual cortex of cats, the CNN model is a specially designed artificial neural network with multiple hidden layers through the biomimetic brain skin layer. Convolution operation is used to solve the disadvantages of large computation and loss of structure information of artificial neural networks. In 1982, Fukushima et al. [21] proposed the concept of Neocognitron to simulate human visual cognitive function, with it being considered to be the starting point of CNNs. In 1989, LeCun et al. [25] built the original Le-Net model, which included a convolutional layer and a fully connected layer. In 1998, LeCun et al. improved and proposed the classical Lenet-5 model, which better solved the problem of handwritten digit recognition.

The birth of LeNet-5 established the basic embryonic form of CNN, which is composed of a convolution layer, a pooling layer, an activation function, and a fully connected layer connected in a certain number of sequential connections, as shown in Figure 3. The CNN is mainly applied to image classification [39–41], object detection [42,43], and semantic segmentation [44–46], as well as in other fields. The most common algorithms are YOLO and R-CNN, among which YOLO has a faster recognition speed due to the characteristics of the algorithm. It has been upgraded to V7. R-CNN's target location search and identification algorithm are slightly different from those of YOLO. Although the speed is slower than that of YOLO, the accuracy rate is higher than that of YOLO.

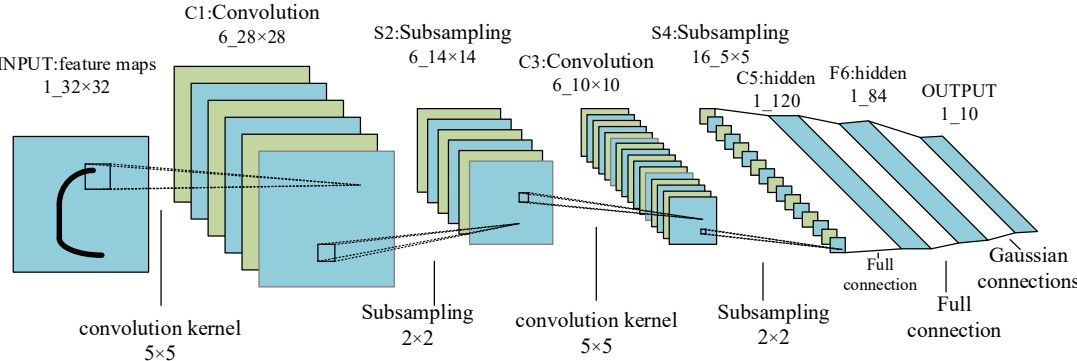

**Figure 3.** The Lenet-5 model.

A convolution neural network with local awareness and parameters share two characteristics, local awareness, namely, the convolution neural network proposed that each neuron need not sense all pixels in the image and local pixels in an image-only perception; then, at a higher level, the information of these local pixels merge, and all the information of the image is obtained. The neural units of different layers are connected locally, that is, the neural units of each layer are only connected with part of the neural units of the previous layer. Each neural unit responds only to the area within the receptive field and does not consider the area outside the receptive field at all. Such a local connection pattern ensures that the learned convolution has the strongest response to the spatial local pattern of the input. The structure of the weight-sharing network makes it more similar to a biological neural network, which reduces the complexity of the network model and reduces the number of weights. This network structure is highly invariant to translation, scaling, tilting, or other forms of deformation. In addition, the convolutional neural network adopts the original image as input, being able to effectively learn the corresponding features from a large number of samples and to avoid the complex feature extraction process.

Since the convolutional neural network can directly process two-dimensional images, it has been widely used in image processing, and many research achievements have been made. The network extracts more abstract features from the original images through simple nonlinear models and only requires a small amount of human involvement in the whole process. However, as each layer of signals can only propagate in one direction and the sample processing is independent of each other at each moment, neither the DNN nor CNN can model the changes in time series. However, in natural language processing, speech recognition, handwriting recognition, and other fields, the chronological order of samples is very critical, and neither DNN nor CNN can deal with these scenarios. Thus, the recurrent neural network (RNN) came into being.

*2.3. Recurrent Neural Network (RNN)*

Different from CNN, RNN introduces the dimension of "time", which is suitable for processing time-series-type data. Because the network itself has a memory ability, it can learn data types with correlation before and after. Recurrent neural networks have a strong model fitting ability for serialized data and are widely used in the field of natural language processing (NLP), including image classification, image acquisition, machine translation, video processing, sentiment analysis, and text similarity calculation. The specific structure is as follows: the recurrent neural network will store and remember the previous information in the hidden layer and then input it into the current calculation of the hidden layer unit.

Figure 4 shows the typical structure of an RNN, which is similar to but different from the traditional deep neural network (DNN). The similarity lies in that the network models of DNN and RNN are fully connected from the input layer to the hidden layer and then

to the output layer, and the network propagation is also sequential. The difference is that the internal nodes of the hidden layer of the RNN are no longer independent of each other but have messages passing to each other. The input of the hidden layer can be composed of the output of the input layer and the output of the hidden layer at a previous time, which indicates that the nodes in the hidden layer are self-connected. It can also be composed of the output of the input layer, the output of the hidden layer at the previous moment, and the state of the previous hidden layer, which indicates that the nodes in the hidden layer are not only self-connected but also interconnected.

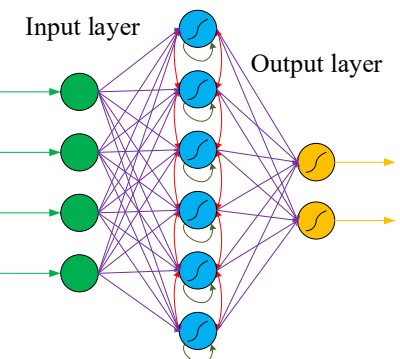

**Figure 4.** Typical structure of an RNN.

Although the RNN solves the problems that the CNN cannot handle, it still has some shortcomings. Therefore, there are many deformed networks of RNN, among which one of the most commonly used networks is the long short-term network (LSTM). The input data of such networks is not limited to images or text, and the problem solved is not limited to translation or text comprehension. Numerical data can also be analyzed using the LSTM. For example, in predictive maintenance applications of factory machines, LSTM can be used to analyze machine vibration signals to predict whether the machine is faulty. In medicine, the LSTM can help in reading through thousands of pieces of literature and can find information related to specific cancers, such as tumor location, tumor size, number of stages, and even treatment policy or survival rate. It can also be combined with image recognition to provide keywords of lesions in order to assist doctors in writing pathological reports.

In addition to the DNN, CNN, and RNN, there is also an emerging network called reinforcement learning, among which the generative adversarial network (GAN) is a distinctive network.

*2.4. Generative Adversarial Network (GAN)*

The GAN (generative adversarial network) is a machine learning model designed by Goodfellow et al. in 2014. Inspired by the zero-sum game in game theory, this model views the generation problem as a competition between generators and discriminators. The generative adversarial network consists of a generator and a discriminator. The generator generates data by learning, and the discriminator determines whether the input data are real data or generated data. After several iterations, the ability of the generator and discriminator is constantly improved, and finally the generated data are infinitely close to the real data so that the discriminator cannot judge whether they are true or false. The GAN is shown in Figure 5.

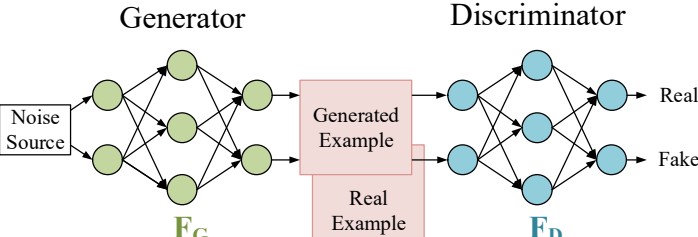

**Figure 5.** Typical structure of a GAN.

The training of the network can be equivalent to the minimax problem of the objective function. That is, let the discriminator maximize the accuracy of distinguishing real data from forged data, and at the same time minimize the probability that the data generated by the generator will be discovered by the discriminator. During training, one of them (the discriminator or generator) is fixed, the parameters of the other model are updated, and the model that can estimate the distribution of sample data is generated by alternating iterations. In GAN, the two networks compete with each other, eventually reaching a balance wherein the generating network can generate data, while the discriminating network can hardly distinguish it from the actual image.

In recent years, with the continuous development of neural networks, the combination of FPGA and neural networks has become increasingly closer, being used in various fields and having achieved certain results.

## 3. Application of Neural Networks Based on FPGA

The largest proportion of FPGA applications is in the field of communication, often used in large flow data transmission, digital signal processing, and other occurrences. With the continuous development of neural networks, the application field of FPGA has expanded from the original communication to a wider range of fields, such as national defense, the military industry, the aerospace industry, industrial control, security monitoring, intelligent medical treatment, and intelligent terminals, and increasingly more intelligent products have been derived from them.

In the academic community, the application of the combination of FPGA and neural networks has attracted increasingly more attention, and the optimal design of neural networks based on FPGA has also become a research hotspot. Scholars have also conducted much research and have obtained some results. We introduce the current research results of the combination of FPGA and neural networks in terms of three aspects: the application of FPGA-based CNN, the application of FPGA-based RNN, and the application of FPGA-based GAN, and analyzed the directions for improvement of these studies.

### 3.1. Application of CNNs Based on FPGA

In terms of intelligent medical treatment, in November 2019, Serkan Sağlam et al. [47] deployed a CNN on FPGA to classify malaria disease cells and achieved 94.76% accuracy. In 2020, Qiang Zhang et al. [48] adopted a CPU+FPGA heterogeneous system to realize CNN classification of heart sound sample data, with an accuracy of 86%. The fastest time to classify 200 heart sound samples was 0.77 s, which was used in the machine of primary hospital to assist in the initial diagnosis of congenital heart disease. In 2021, Jiying Zhu et al. [49] proposed a computed tomography (CT) diagnostic image recognition of cholangiocarcinoma that was based on FPGA and a neural network, showing an excellent classification performance on liver dynamic CT images. In September of the same year, Siyu Xiong et al. [50] used an FPGA-based CNN to speed up the detection and segmentation of 3D brain tumors, providing a new direction for the improvement of automatic segmentation of brain tumors. In 2022, H. Liu et al. [51] designed an FPGA-based multi-task recurrent neural network gesture recognition and motion assessment upper limb

rehabilitation device, making it unnecessary for patients with upper limb movement disorders to spend a large amount of time in the hospital for upper limb rehabilitation training. Using this device can help patients to complete the same professional upper limb rehabilitation training at home in the same way as in the hospital, helping the patients in reducing the burden of medical expenses and reducing the investment of a large number of medical resources. The intelligent rehabilitation system adopts a high-level synthesis design of HLS, and a convolutional recurrent neural network (C-RNN) is deployed on Zynq ZCU104 FPGA. The experiments showed that the system can recognize the unnatural features (such as tremor or limited flexion and extension) in patients' dynamic movements and upper limb movements with an accuracy of more than 99%. However, the system does not optimize the convolution operation in the network, nor does it optimize the memory access of parameters.

In terms of national defense, in 2020, Cihang Wang et al. [52] found that a large number of high-precision and high-resolution remote sensing images were used in civil economic construction and military national defense, and thus they proposed a scheme of real-time processing of remote sensing images that was based on a convolutional neural network on the FPGA platform. This scheme optimized CNN on FPGA from two aspects: spatial parallel and temporal parallel. By adding pipeline design and using the ideas of module reuse and data reuse, the pressure of the data cache was reduced, and the resource utilization of FPGA was increased. Compared with other schemes, the proposed scheme greatly improved the recognition speed of remote sensing images, reduced the power consumption, and achieved 97.8% recognition accuracy. Although the scheme uses many speedup techniques, it does not optimize the convolution operation, which has the most significant performance improvement. In the next step, the traditional CNN convolution operation can be transformed into a Winograd fast convolution operation, so as to reduce the use of multiplication and accumulation operation, and thus improving the model operation rate and reducing the resource occupancy.

In the same year, Buyue Qin et al. [53] proposed a special processor for key point detection of aircraft that was based on FPGA and deployed a VGG-19 deep neural network (DNN) to speed up the detection process of enemy aircraft, so as to detect enemy aircraft in the first time and avoid the danger of being attacked. The design was implemented on Xilinx Virtex-7 VC709 FPGA at 150 MHz (Mega Hertz) using HLS high-level synthesis, fixed-point quantization, on-chip data buffering, and FIFO (first in first out) optimization methods. Compared to the Intel I7-8700K (@ 3.7 GHz, Giga Hertz) processor, the former is 2.95 times the throughput of the latter and 17.75 times the performance power ratio. Although the processor uses HLS high-level synthesis to simplify the difficulty of network deployment on FPGA, HLS causes individuals to focus on the design and to pay less attention to the specific implementation of the bottom layer by integrating other languages such as C/C++ (the C/C++ programming language) into the HDL (hardware description language). Compared with the implementation of VHDL (very high-speed integrated circuit hardware description language) or Verilog (Verilog HDL), the optimization objective of the algorithm is not the actual on-board objective, and thus the final implementation often fails to meet the timing or power constraints.

### 3.2. Application of RNNs Based on FPGA

At present, in addition to the application of deep neural networks into the fields of image and video, an FPGA-based speech recognition system has also become a research hotspot. Among them, the commonly used speech recognition model is the RNN and its variant LSTM. Due to its huge market demand, speech recognition develops rapidly. In intelligent speech recognition products, in order to ensure certain flexibility and mobility, the speech recognition model is usually deployed on FPGA to meet the needs of intelligence and production landing.

In 2016, the work of J. C. Ferreira and J. Fonseca et al. [54] was considered as one of the earliest works to implement the LSTM network on the FPGA hardware platform,

which focused people's attention from the software implementation of LSTM to FPGA hardware. In 2017, Y. Guan et al. [55] implemented the LSTM network using special hardware IP containing devices such as data scheduler, FPGA, AXI4Lite (one of the Advanced eXtensible Interfaces) bus, and optimized computational performance and communication requirements in order to accelerate inference. In the same year, Y. Zhang et al. conducted two works. First, they proposed an energy-efficient optimization structure based on tiling vector multiplication, binary addition tree, overlap computation, and data access methods [56]. The second further developed on the basis of the first work and improved the performance by adding sparse LSTM layers that occupy less resources [57]. Song Han et al. [58] proposed the ESE (efficient speech recognition engine), a sparse LSTM efficient speech recognition engine based on FPGA, which uses a load-balancing sensing pruning method. The proposed method can compress the size of the LSTM model by a factor of 20 (10 times pruning, 2 times quantization) with negligible loss of prediction accuracy. They also proposed a scheduler that encodes and partitions the compression model into multiple parallel PE and schedule complex LSTM data streams. Compared with the peak performance of the uncompressed LSTM model of 2.52 GOPS (Giga Operations Per Second), the ESE engine reached the peak performance of 282 GOPS on a Xilinx XCKU060 FPGA with 200 MHz operating frequency. ESE, evaluated in the LSTM speech recognition benchmark, was 43 and 3 times faster than the Intel Core I7 5930K CPU and Pascal Titan X GPU implementations, respectively, and was 40 and 11.5 times more energy efficient, respectively.

In 2018, Zhe Li et al. [59] summarized two previous works on the FPGA implementation of the LSTM RNN inference stage on the basis of model compression. One work found that the network structure became irregular after weight pruning. Another involved adding a cyclic matrix to the RNN to represent the weight matrix in order to achieve model compression and acceleration while reducing the influence of irregular networks. On this basis, an efficient RNN framework for automatic speech recognition based on FPGA was proposed, called E-RNN. Compared with the previous work, E-RNN achieved a maximum energy efficiency improvement of 37.4 times, which was more than two times that of the latter work under the same accuracy. In 2019, Yong Zheng et al. [60] also used the pruning operation to compress the LSTM model, which was different from the work of Zhe Li et al. [59] in that they used the permutation block diagonal mask matrix to pry the model. The structured sparse features were created, and the normalized linear quantization method was used to quantify the weight and activation function, so as to achieve the purpose of hardware friendliness. However, compared with similar jobs, the acceleration effect was not very significant.

In 2020, Yuxi Sun et al. [61] adopted the idea of model partitioning. Unlike Song Han et al. [58], who coded the compression model and divided it into multiple parallel PEs for processing, they extended the reasoning of deep RNN by dividing a large model into FPGA clusters. The whole FPGA cluster shared an RNN model, and thus each FPGA processed only a part of the large RNN model, which reduced the processing burden of FPGA. The parallelism of the FPGA cluster and the time dependence of RNN made the delay basically unchanged when the number of RNN layers increased. Compared to Intel CPU, 31 times and 61 times speedup were achieved for single-layer and four-layer RNNS, respectively.

In December of the same year, Chang Gao et al. [62] proposed a lightweight gated recursive unit (GRU)-based RNN accelerator, called EDGERDNN, which was optimized for low-latency edge RNN inference with batch size 1. EDGERDRNN used a delta network algorithm inspired by pulsed neural networks in order to exploit the temporal sparsity in RNN. Sparse updates reduced distributed RAM (DRAM) weight and memory access by a factor of 10, and reduced latency, resulting in an average effective throughput of 20.2 GOP/s for batch size 1. In 2021, Jinwon Kim et al. [63] implemented an efficient and reconfigurable RNN inference processor AERO on a resource-limited Intel Cyclone V FPGA on the basis of the instruction set architecture that specializes in processing the raw

vector operations that constitute the data stream of RNN models. The vector processing unit (VPU) based on the approximation multiplier was used to complete the approximate calculation in order to reduce the resource usage. Finally, AERO achieved a resource utilization of 1.28 MOP/s/LUT. As the authors stated, the next step can be achieved by the multi-core advantage of FPGA clusters in order to achieve higher reasoning speed.

In June 2022, Jianfei Jiang et al. [64] proposed a subgraph segmentation scheme based on the CPU-FPGA heterogeneous acceleration system and the CNN-RNN hybrid neural network. The Winograd algorithm was applied to accelerate the CNN. Fixed-point quantization, cyclic tiling, and piecewise linear approximation of activation function were used to reduce hardware resource usage and to achieve high parallelization in order to achieve RNN acceleration. The connectionist text proposal network (CTPN) was used for testing on an Intel Xeon 4116 CPU and Arria10 GX1150 FPGA, and the throughput reached 1223.53 GOP/s. Almost at the same time, Chang Gao et al. [65] proposed a new LSTM accelerator called "Spartus" that was again based on spatiotemporal sparsity, with it utilizing spatiotemporal sparsity in order to achieve an ultra-low delay inference. Different from the pruning method used by Zhe Li [59] and Yong Zheng et al. [60], they used a new structured pruning method of column balancing target dropout (CBTD) in order to induce spatial sparsity, which generates structured sparse weight matrices for balanced workloads and reduces the weight of memory access and related arithmetic operations. The throughput of 9.4 TOp/s and energy efficiency of 1.1 Top/J were achieved on a Xilinx Zynq 7100 FPGA with an operating frequency of 200 MHz.

### 3.3. Application of GANs Based on FPGA

In 2018, Amir Yazdanbakhsh et al. [66] found that generators and discriminators in GANs use convolution operators differently. The discriminator uses the normal convolution operator, while the generator uses the transposed convolution operator. They found that due to the algorithmic nature of transposed convolution and the inherent irregularity in its computation, the use of conventional convolution accelerators for GAN leads to inefficiency and underutilization of resources. To solve these problems, FlexiGAN was designed, an end-to-end solution that generates optimized synthesizable FPGA accelerators according to the advanced GAN specification. The architecture takes advantage of MIMD (multiple instructions stream multiple data stream) and SIMD (single instruction multiple data) execution models in order to avoid inefficient operations while significantly reducing on-chip memory usage. The experimental results showed that FlexiGAN produced an accelerator with an average performance of 2.2 times that of the optimized conventional accelerator. The accelerator delivered an average of 2.6 times better performance per watt compared to the Titan X GPU. Along the same lines as Amir Yazdanbakhsh et al., Jung-Woo Chang et al. [67] found that the GAN created impressive data mainly through a new type of operator called deconvolution or transposed convolution. On this basis, a Winograd-based deconvolution accelerator wsa proposed, which greatly reduces the use of the multiplication–accumulation operation and improves the operation speed of the model. The acceleration effect was 1.78~8.38 times of the fastest model at that time. However, as was the case for Amir Yazdanbakhsh et al., they only optimized the convolution operation in GANs, but did not optimize the GAN model framework.

In the study of infrared image colorization, in order to obtain more realistic and detailed colorization results, in 2019, Xingping Shi et al. [68] improved the generator based on Unet, designed a discriminator for deconvolution optimization, proposed a DenseUnet GAN structure, and added a variety of loss functions in order to optimize the colorization results. The data were preprocessed, and the face localization neural network used in preprocessing datasets was accelerated by FPGA. It achieved better results than other image coloring methods on large public datasets.

In terms of image reconstruction research, Dimitrios Danopoulos et al. [69] first used GAN to implement image reconstruction application on FPGA in 2021. Compared with CPU and GPU platforms, generator models trained with specific hardware optimizations

can reconstruct images with high quality, minimum latency, and the lowest power consumption.

In terms of hardware implementation of GAN, in 2021, Yue Liu et al. [70] proposed a hardware scheme of a generative adversarial network based on FPGA, which effectively meets the requirements of dense data communication, frequent memory access, and complex data operation in generative adversarial networks. However, the parameters of GANs are not optimized by the acceleration design such as ping-pong cache, and thus there is a large amount of room for improvement in this research.

## 4. Neural Network Optimization Technology Based on FPGA

With the rapid development of artificial intelligence, the number of layers and nodes of neural network models is increasing, and the complexity of the models is also increasing. Deep learning and neural networks have put forward more stringent requirements on the computing ability of hardware. On the basis of the advantages of FPGA mentioned in the introduction, increasingly more scholars are choosing to use FPGA to complete the deployment of neural networks. As shown in Figure 6, according to different design concepts and requirements, FPGA-based neural network optimization technology can be roughly divided into optimization for data and operation, optimization for bandwidth, and optimization for memory and access, among others, which are introduced in detail below.

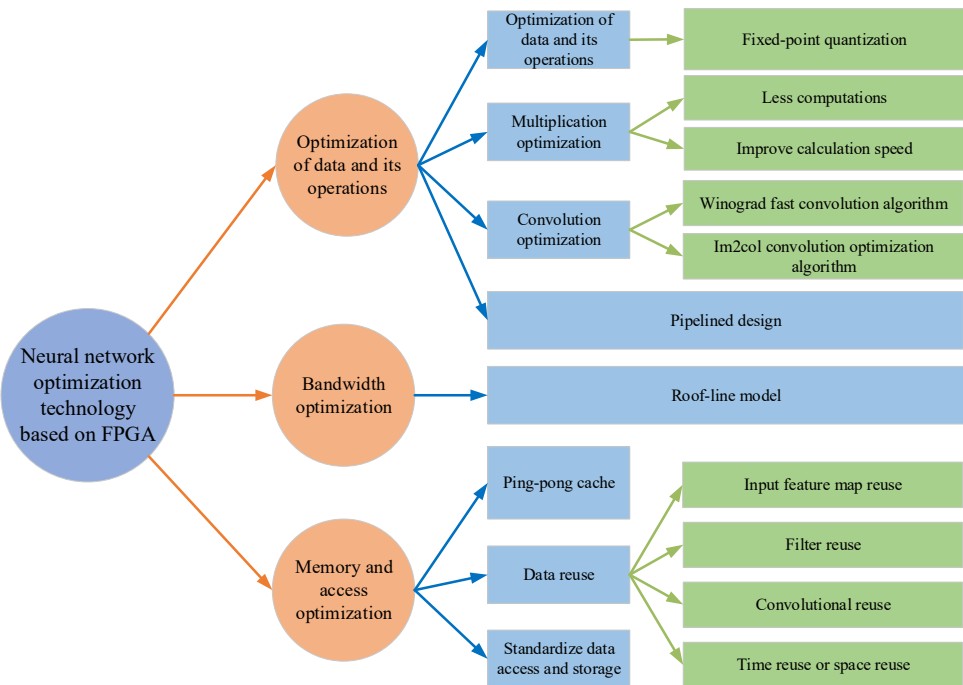

**Figure 6.** Neural network optimization technology based on FPGA.(fixed-point quantization [71–78], less computations [79–81], improve calculation speed [82–85], Winograd fast convolution algorithm [86–91], Im2col convolution optimization algorithm [92–97], pipelined design [98–102], Roofline model [103–105], ping-pong cache [106–109], input feature map reuse [110,111], filter reuse [111,112], convolutional reuse [110–112], time reuse or space reuse [111], standardize data access and storage [113–115]).

### 4.1. Optimization of Data and Its Operations

In the aspect of optimization of data and its operation, scholars have made many attempts and achieved certain results. Aiming at the data itself, a method to reduce the data accuracy and computational complexity is usually used. For example, fixed-point

quantization is used to reduce the computational complexity with an acceptable loss of data accuracy, so as to improve the computational speed. In terms of operation, the optimization is realized by reducing the computation times of multiplication and increasing the computation speed. The commonly used optimization methods include addition calculation instead of multiplication calculation, the Winograd fast convolution algorithm, and the Im2col convolution acceleration algorithm, among others.

### 4.1.1. Fixed-Point Quantitative

Since the bit width of the number involved in the operation is finite and constant in FPGA calculations, the floating-point decimal number involved in the operation needs to be fixed to limit the bit width of a floating-point decimal to the range of bit width allowed by FPGA. Floating-point decimals mean that the decimal point position is not fixed, and fixed-point decimals mean that the decimal point position is fixed. Fixed-point quantization is the use of finite digits to represent infinite precision numbers, that is, the quantization function maps the full precision numbers (the activation parameters, weight parameters, and even gradient values) to a finite integer space. Fixed-point quantization can greatly reduce the memory space of each parameter and the computational complexity within the acceptable accuracy loss range, so as to achieve neural network acceleration.

In 2011, Vincent Vanhoucke first proposed the linear fixed-point 8-bit quantization technology, which was initially used in X86 CPU, greatly reducing the computational cost, with it then being slowly applied to FPGA [71]. In March 2020, Shiguang Zhang et al. [72] proposed a reconfigurable CNN accelerator with an AXI bus based on advanced RISC machine (ARM)+FPGA architecture. The accelerator receives the configuration signal sent by the ARM. The calculation in the reasoning process of different CNN layers is completed by time-sharing, and the data movement of the convolutional layer and pooling layer is reduced by combining convolutional and pooling operations; moreover, the number of access times of off-chip memory is reduced. At the same time, fixed-point quantization is used to convert floating-point numbers into 16-bit dynamic fixed-point format, which improves the computational performance. The peak performance of 289 GOPS is achieved on a Xilinx ZCU102 FPGA.

In June of the same year, Zongling Li et al. [73] designed a CNN weight parameter quantization method suitable for FPGA, different from the direct quantization operation in the literature [72]. They transformed the weight parameter into logarithm base 2, which greatly reduced the quantization bit width, improved the quantization efficiency, and reduced the delay. Using a shift operation instead of a convolution multiplication operation saves a large number of computational resources. In December of the same year, Sung-en Chang et al. [74] first proposed a hardware-friendly quantization method named Sum-of-Power-of-2 (SP2), and on the basis of this method proposed a mixed scheme quantization (MSQ) combining SP2 and fixed-point quantization methods. By combining these two schemes, a better match with the weight distribution is achieved, accomplishing the effect of maintaining or even improving the accuracy.

In 2021, consistent with X. Zhao's view of whole-integer quantization [75], Zhenshan Bao et al. [76] proposed an effective quantization method that was based on hardware implementation, a learnable parameter soft clipping fully integer quantization (LSFQ). Different from previous studies that only quantified weight parameters, input and output data, etc., their study quantified the entire neural network as integers and automatically optimized the quantized parameters in order to minimize losses through backpropagation. Then, the batch norm layer and convolution layer were fused to further quantify the deviation and quantization step. In the same year, Xiaodong Zhao et al. [77] optimized the YOLOv3 network structure through pruning and the Int8 (Integer8) quantization algorithm with the trade-off between speed and accuracy. Good acceleration was achieved with limited and acceptable loss of accuracy. Some scholars proposed the use of the quantization algorithm to map matrix elements participating in matrix multiplication operations from single-precision floating-point data to half-precision floating-point data. Under

the condition of ensuring the accuracy of the algorithm model, the consumption of on-chip storage resources on FPGA is greatly saved and the computing speed is improved [78].

### 4.1.2. Multiplication Optimization

Matrix operations mainly appear in the training and forward computation of neural networks and they occupy a dominant position; thus, it is of great significance to accelerate matrix operation. The optimization of matrix multiplication can be achieved by reducing the number of multiplication computations and increasing the computation speed. Matrix operations often have remarkable parallelism. Scholars usually use the characteristics of FPGA parallel computing to optimize matrix operations. In 2006, D. K. Iakovidis et al. [82] designed an FPGA structure capable of performing fast parallel co-occurrence matrix computation in grayscale images. The symmetries and sparsity of a co-occurrence matrix were used to achieve shorter processing time and less FPGA resource occupation. In 2014, Jeremy Fowers et al. [79] designed an FPGA accelerator with high memory bandwidth for Sparse matrix–vector multiplication.

As shown in Figure 7, the architecture uses specialized compressed interleave sparse row (CISR) coding to efficiently process multiple rows of the matrix in parallel, combined with a caching design that eliminates the replication of the buffer carrier and enables larger vectors to be stored on the chip. The design maximizes the bandwidth utilization by organizing the data from memory into parallel channels, which can keep the hardware complexity low while greatly improving the parallelism and speed of data processing.

In 2016, Eriko Nurvitadhi et al. [80] used XNOR Gate to replace the multiplier in FPGA in order to reduce the computational difficulty and improve the computational efficiency from the perspective of the multiplier occupying more resources. In 2019, Asgar Abbaszadeh et al. [83] proposed a universal square matrix computing unit that was based on cyclic matrix structure and finally tested a 500 × 500 matrix on an FPGA with an operating frequency of 346 MHz, achieving a throughput of 173 GOPS. In 2020, S. Kala and S. Nalesh [84] proposed an efficient CNN accelerator that was based on block Winograd GEMM (general matrix multiplication) architecture. Using blocking technology to improve bandwidth and storage efficiency, the ResNet-18 CNN model was implemented on XC7VX690T FPGA. Running at a clock frequency of 200MHz, the average throughput was 383 GOPS, which was a significant improvement in comparison to the work of Asgar Abbaszadeh et al. In the same year, Ankit Gupta et al. [81] made a tradeoff between accuracy and performance and proposed a new approximate matrix multiplier structure, which greatly improved the speed of matrix multiplication by introducing a negligible error amount and an approximate multiplication operation.

In 2022, Shin-Haeng Kang et al. [85] implemented the RNN-T (RNN-Transducer) inference accelerator on FPGA for the first time on the basis of Samsung high-bandwidth memory–processing in memory (HBM-PIM) technology. By placing the DRAM-optimized AI engine in each memory bank (storage subunit), the processing power was brought directly to the location of the data storage, which enabled parallel processing and minimized data movement. The HBM internal bandwidth was utilized to significantly reduce the execution time of matrix multiplication, thus achieving acceleration.

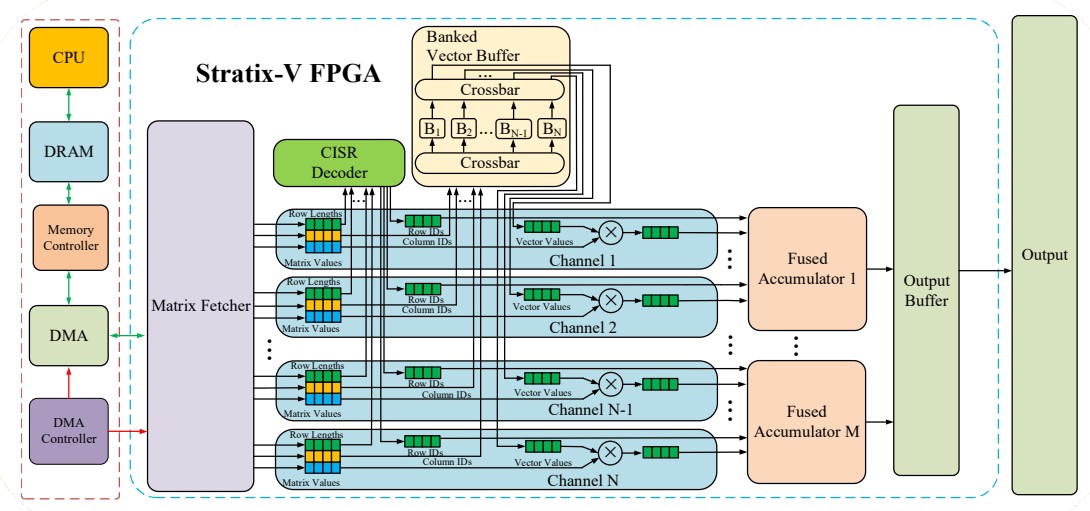

**Figure 7.** Neural network optimization technology based on FPGA.

### 4.1.3. Convolution Optimization

For the optimization of convolution algorithms, scholars have provided the following two ideas. One is the Winograd fast convolution algorithm, which speeds up convolution by increasing addition and reducing multiplication. Another is the Im2col convolution optimization algorithm that sacrifices storage space in order to improve the speed of the convolution operation. This is described in detail below.

(1)  Winograd fast convolution algorithm

The Winograd Fast Convolutional algorithm was first proposed by Shmuel Winograd in his paper "Fast Algorithms for Convolutional Neural Networks" in 1980, but it did not cause much sensation at that time. In the 2016 CVPR conference, Lavin et al. [86] proposed the use of the Winograd fast convolution algorithm to accelerate the convolution operation. Since then, the Winograd algorithm has been widely used to accelerate the convolution algorithm.

The Winograd algorithm can accelerate the convolution operation because it uses more addition to reduce multiplication calculation, thus reducing the amount of calculations, using less FPGA resources, and improving operation speed, not as well as when FFT (fast Fourier transform) is introduced into the plural. But with the premise being, in the processor, the clock cycle of the calculation of the multiplication are longer than the clock cycles of the addition.

Taking the one-dimensional convolution operation as an example, the input signal as $d = [d_0\ d_1\ d_2\ d_3]^T$, and the convolution kernel as $g = [g_0\ g_1\ g_2]^T$, then the convolution can be written as the following matrix multiplication form:

$$F(2,3) = \begin{pmatrix} d_0 & d_1 & d_2 \\ d_1 & d_2 & d_3 \end{pmatrix} \begin{pmatrix} g_0 \\ g_1 \\ g_2 \end{pmatrix} = \begin{pmatrix} r_0 \\ r_1 \end{pmatrix} \tag{1}$$

For general matrix multiplication, six multiplications and four additions are required, as follows: $r_0 = d_0 g_0 + d_1 g_1 + d_2 g_2$, $r_1 = d_1 g_0 + d_2 g_1 + d_3 g_2$. However, the matrix transformed by the input signal in the convolution operation is not an arbitrary matrix, in which a large number of repetitive elements are regularly distributed, such as $d_1$ and $d_2$ in row 1 and row 2. The problem domain of the matrix multiplication transformed by convolution is smaller than that of the general matrix multiplication, which makes it possible to optimize. After conversion using a Winograd algorithm:

$$F(2,3) = \begin{pmatrix} d_0 & d_1 & d_2 \\ d_1 & d_2 & d_3 \end{pmatrix} \begin{pmatrix} g_0 \\ g_1 \\ g_2 \end{pmatrix} = \begin{pmatrix} m_1 + m_2 + m_3 \\ m_2 - m_3 - m_4 \end{pmatrix} \tag{2}$$

Among them, the $m_1 = (d_0 - d_2) g_0$, $m_2 = (d_1 + d_2) (g_0 + g_1 + g_2)/2$, $m_3 = (d_2 - d_1) (g_0 - g_1 + g_2)/2$, $m_4 = (d_1 - d_3) g_2$. To calculate $r_0 = m_1 + m_2 + m_3$, $r_1 = m_2 - m_3 - m_4$, the number of operations required are four additions (subtraction) on the input signal d and four multiplications and four additions on the output m, respectively. In the inference stage of the neural network, the elements on the convolution kernel are fixed, and therefore the operation on g can be calculated well in advance. In the prediction stage, it only needs to be calculated once, which can be ignored. Therefore, the total number of operations required is the sum of the operation times on *d* and *m*, namely, four times of multiplication and eight times of addition. Compared with the direct operation of six multiplications and four additions, the number of multiplications decreases, and the number of additions increases. In FPGA, the multiplication operation is much slower than the addition operation, and it will occupy more resources. By reducing multiplication times and adding a small amount of addition, the operation speed can be improved.

It is worth noting that the Winograd algorithm is not a panacea. As the number of additions increases, additional transform computation and transform matrix storage are required. As the size of convolution kernel and tile increases, the cost of addition, transform, and storage needs to be considered. Moreover, the larger the tile, the larger the transform matrix, and the loss of calculation accuracy will further increase. Therefore, the general Winograd algorithm is only suitable for small convolution kernels and tiles.

In FPGA, the neural network optimization based on the Winograd algorithm also has a large number of research achievements. In 2018, Liqiang Lu et al. [87] proposed an efficient sparse Winograd convolutional neural network accelerator (SpWA) that was based on FPGA. Using the Winograd fast convolution algorithm, transforming feature maps to specific domains to reduce algorithm complexity, and compressing CNN models by pruning unimportant connections were able to reduce storage and arithmetic complexity. On the Xilinx ZC706 FPGA platform, at least 2.9× speedup was achieved compared to previous work. In 2019, Kala S. et al. [88] combined the Winograd algorithm and GEMM on FPGA to speed up the reasoning of AlexNet. In 2020, Chun Bao et al. [89] implemented the YOLO target detection model based on the Winograd algorithm under PYNQ (Python productivity for Zynq) architecture, which was applied in order to accelerate edge computing and greatly reduced the resource usage of FPGA and the power consumption of the model. In November of the same year, Xuan Wang et al. [90] systematically analyzed the challenges of simultaneously supporting the Winograd algorithm, weight sparsity, and activation sparsity. An efficient encoding scheme was proposed to minimize the effect of activation sparsity, and a new decentralized computing-aggregation method was proposed to deal with the irregularity of sparse data. It was deployed and evaluated on Zed-Board, ZC706, and VCU108 platforms and achieved the highest energy efficiency and DSP (digital signal processing) efficiency at the time. In 2022, Bin Li et al. [91] combined the Winograd algorithm with fusion strategy, reduced the amount of data movement and the number of accesses to off-chip memory, and improved the overall performance of the accelerator. On the U280 FPGA board, the mean average precision (mAP) decreased by 0.96% after neural network quantization, and the performance reached 249.65 GOP/s, which was 4.4 times Xilinx's official parameters.

(2)　Im2col convolution optimization algorithm

The Im2col convolution optimization algorithm is different from the Winograd algorithm, which uses more addition operations instead of multiplication operations in order to occupy less resources in order to improve the operation speed. The Im2col convolution optimization algorithm adopts the idea of exchanging space for the improvement of computation speed. By converting the receptive field of convolution kernel into a row

(column) for storage, the memory access time is reduced, and the operation speed is accelerated.

Because matrix multiplication involves nothing more than the dot product of the corresponding position, it only takes one traversal for both matrices to be multiplied, and thus matrix multiplication is very fast compared to convolution. From the perspective of matrix multiplication, the convolution operation is actually the process of circular matrix multiplication with the template moving. Although the image of each multiplication is locally different, the template remains the same. Every time one adds a template and a local dot product, one multiplies rows and columns in matrix multiplication.

As shown in Figure 8, using the Im2col convolution optimization algorithm as an example, by converting each local multiplication and accumulation process into the multiplication of a row and a column of two matrices, the whole convolution process can be converted into a matrix multiplication process, and the convolution speed can be greatly improved. At the same time, it is worth noting that just because of the action mechanism of Im2col convolution optimization algorithm, the number of elements after Im2col expansion will be more than the number of elements of the original block. Therefore, optimizing the convolution operation using Im2col consumes more memory. Therefore, when the Im2col convolution optimization algorithm is used, memory optimization technology is often used together.

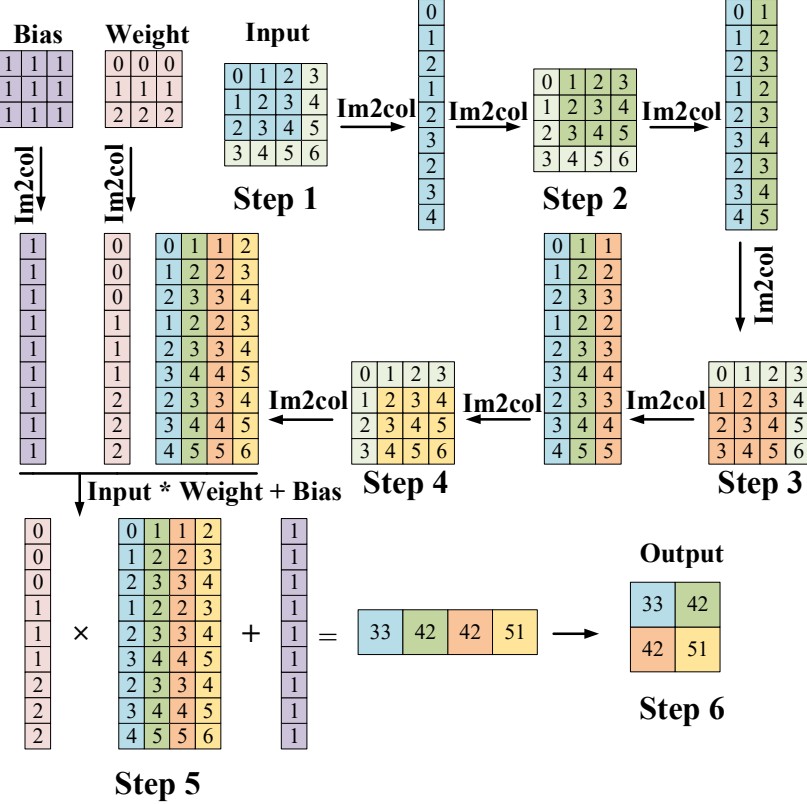

**Figure 8.** An example of the Im2col convolution optimization algorithm.

In terms of FPGA-based neural network Im2col convolution optimization, in 2017, Feixue Tang et al. [92] used the Im2col algorithm to optimize the convolution algorithm and then converted the optimized convolution algorithm into a matrix operation, so as to improve the operation speed. In 2020, Feng Yu et al. [93] combined the quantization method with the Im2col algorithm for visual tasks and designed a dedicated data-stream convolution acceleration under the heterogeneous CPU-FPGA platform PYNQ, including

the changed data-stream order and Im2col for convolution. Compared with ARM CPU, 120× acceleration was achieved on FPGA operating at 250 MHz.

In 2021, Tian Ye et al. [94] applied the Im2col algorithm to the FPGA accelerator of homomorphic encryption of private data in order to optimize convolution, which greatly reduced the delay of the convolution algorithm. In October of the same year, Hongbo Zhang et al. [95] combined the Im2col algorithm with pipeline design and applied it to the lightweight object detection network YOLOV3-Tiny, achieving 24.32 GOPS throughput and 3.36 W power consumption on the Zedboard FPGA platform.

In 2022, Chunhua Xiao et al. [96] proposed a smart data stream transformation method (SDST), which is similar to but different from the Im2col algorithm. Different from the Im2col algorithm, with the improvement of the peak performance of the GEMM kernel, the overhead caused by the Im2col algorithm, which converts convolution to matrix multiplication, becomes obvious. SDST divides the input data into conflict-free streams on the basis of the local property of data redundancy, which reduces the external storage of data and keeps the continuity of data. A prototype system based on SDST was implemented on Xilinx ZC706 APSoC (all programmable system-on-chip), and the actual performance of accelerated CNN was evaluated on it. The results showed that SDST was able to significantly reduce the overhead associated with the explicit data conversion of convolutional inputs at each layer. In March of the same year, Bahadır Özkılbaç et al. [97] combined a fixed-point quantization technique with the Im2col algorithm in order to optimize convolution, reduce temporary memory size and storage delay, perform the application of digital classification of the same CNN on acceleration hardware in the ARM processor and FPGA, and observe the delay time. The results show that the digital classification application executed in the FPGA was 30 times faster than that executed in the ARM processor.

### 4.1.4. Pipelined Design

Pipelined design is a method of systematically dividing combinational logic, inserting registers between the parts (hierarchies), and temporarily storing intermediate data. The purpose is to decompose a large operation into a number of small operations. Each small operation takes less time, shortens the length of the path that a given signal must pass in a clock cycle, and improves the frequency of operations; moreover, small operations can be executed in parallel and can improve the data throughput.

The design method of Pipeline can greatly improve the working speed of the system. This is a typical design approach that converts "area" into "velocity." The "area" here mainly refers to the number of FPGA logic resources occupied by the design, which is measured by the consumed flip-flop (FF) and look-up table (LUT). "Speed" refers to the highest frequency that can be achieved while running stably on the chip. The two indexes of area and speed always run through the design of FPGA, which is the final standard of design quality evaluation. This method can be widely used in all kinds of designs, especially the design of large systems with higher speed requirements. Although pipelining can increase the use of resources, it can reduce the propagation delay between registers and ensure that the system maintains a high system clock speed. In the convolutional layer of deep neural networks, when two adjacent convolutional iterations are performed and there is no data dependency, pipelining allows for the next convolutional operation to start before the current operation is completely finished, improving the computational power, with pipelining having become a necessary operation for most computational engine designs. In practical application, considering the use of resources and the requirements of speed, the series of pipelines can be selected according to the actual situation in order to meet the design needs.

In terms of using pipeline design to accelerate neural networks, pipeline design is usually not used alone, but instead used in conjunction with other techniques. In 2017, Zhiqiang Liu et al. [98] used a pipeline design together with the space exploration method to maximize the throughput of a network model, optimizing and evaluating three

representative CNNs (LeNet, AlexNet, and VGG-S) on a Xilinx VC709 board. The results show that the performances of 424.7 GOPS/s, 445.6 GOPS/s, and 473.4 GOPS/s, respectively, were clearly higher than in previous work.

In 2019, Yu Xing et al. [99] proposed an optimizer that integrates graphics, loops, and data layout, as well as DNNVM, a full-stack compiler for an assembler. In the framework of a compiler, the CNN model is transformed into a directed acyclic graph (XGraph). The pipeline design, data layout optimization, and operation fusion technology are applied to XGraph, and lower FPGA hardware resource consumption is realized on VGG and Residual Network 50 (ResNet50) neural networks. On GoogLeNet with operation fusion, a maximum 1.26× speedup was achieved compared to the original implementation. In April of the same year, Wei Wang et al. [100] combined a pipeline design with a ping-pong cache and optimized Sigmoid activation function through a piecewise fitting method combining the look-up table and polynomial. On a Xilinx virtex-7 FPGA with 150 MHz operating frequency, the computational performance was improved from 15.87 GOPS to 20.62 GOPS, and the recognition accuracy was 98.81% on a MNIST dataset.

In 2021, Dong Wen et al. [101] used the CNN for acoustic tasks and combined fixed-point quantization, space exploration methods similar to the roof-line model, and pipeline design in order to achieve higher throughput of the designed FPGA accelerator. In 2022, Varadharajan et al. [102] proposed pipelined stochastic adaptive distributed architectures (P-SCADAs) by combining pipelined design and adaptive technology in the LSTM network, which improved FPGA performance and saved FPGA resource consumption and power consumption.

## 4.2. Bandwidth Optimization

In the deployment of neural networks, all kinds of neural networks must rely on specific computing platforms (such as CPU/GPU/ASIC/FPGA) in order to complete the corresponding algorithms and functions. At this point, the "level of compatibility" between the model and the computing platform will determine the actual performance of the model. Samuel Williams et al. [103] proposed an easy-to-understand visual performance model, the roof-line model, and proposed a quantitative analysis method using operational intensity. The formula showcasing that the model can reach the upper limit of the theoretical calculation performance on the computing platform is provided. It offers insights for programmers and architects in order to improve the parallelism of floating-point computing on software and hardware platforms and is widely used by scholars to evaluate their designs on various hardware platforms to achieve better design results.

The roof-line model is used to measure the maximum floating point computing speed that the model can achieve within the limits of a computing platform. Computing power and bandwidth are usually used to measure performance on computing platforms. Computing power is also known as the platform performance ceiling, which refers to the number of floating pointed operations per second that can be completed by a computing platform at its best, in terms of FLOP/s or GLOP/s. Bandwidth is the maximum bandwidth of a computing platform. It refers to the amount of memory exchanged per second (byte/s or GB/s) that can be completed by a computing platform at its best. Correspondingly, the computing intensity limit Imax is the computing power divided by the bandwidth. It describes the maximum number of calculations per unit of memory exchanged on the computing platform in terms of FLOP/byte. The force calculation formula of the roof-line model is as follows:

$$P = \begin{cases} \beta \times I, I < I_{\max} \\ \pi, \quad x \geqslant I_{\max} \end{cases} \tag{3}$$

As shown in Figure 9, the so-called "roof-line" refers to the "roof" shape determined by the two parameters of computing power and bandwidth upper limit of the computing platform. The green line segment represents the height of the "roof" determined by the

computing power of the computing platform, and the red line segment represents the slope of the "eaves" determined by the bandwidth of the computing platform.

As can be seen from the figure, the roof-line model is divided into two areas, namely, the computation limited area and the memory-limited area. In addition, we can find three rules from Figure 9:

(1) When the computational intensity of the model I is less than the upper limit of the computational intensity of the computing platform Imax. Since the model is in the "eaves" interval at this time, the theoretical performance P of the model is completely determined by the upper bandwidth limit of the computing platform (the slope of the eaves) and the computational strength I of the model itself. Therefore, the model is said to be in a memory-limited state at this time. It can be seen that under the premise that the model is in the bandwidth bottleneck, the larger the bandwidth of the computing platform (the steeper the eaves), or the larger the computational intensity I of the model, the greater the linear increase in the theoretical performance P of the model.

(2) When the computational intensity of the model I is greater than the upper limit of the computational intensity of the computing platform Imax. In the current computing platform, the model is in the compute-limited state, that is, the theoretical performance P of the model is limited by the computing power of the computing platform and can no longer be proportional to the computing strength I.

(3) No matter how large the computational intensity of the model is, its theoretical performance P can only be equal to the computational power of the computing platform at most.

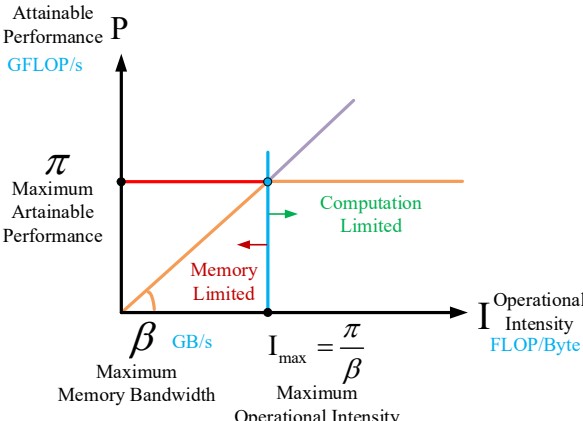

**Figure 9.** Roof-line model.

It can be seen that increasing the bandwidth ceiling or reducing the bandwidth requirement of the system can improve the performance of the system and thus accelerate it. Specifically, the bandwidth requirement of the system can be reduced by the techniques of data quantization and matrix calculation (described in detail in Section 4.1), and the upper limit of the bandwidth can be increased by data reuse and optimization of data access (described in detail in Section 4.3).

In recent years, many application achievements based on the roof-line model are also involved in the optimization design of neural networks that are based on FPGA. In 2020, Marco Siracusa et al. [104] focused on the optimization of high-level synthesis (HLS). They found that although advanced synthesis (HLS) provides a convenient way to write FPGA code in a general high-level language, it requires a large amount of effort and expertise to optimize the final FPGA design of the underlying hardware, and therefore they proposed a semi-automatic performance optimization method that was based on the FPGA-based hierarchical roof line model. By combining the FPGA roof-line model with the Design

Space Exploration (DSE) engine, this method is able to guide the optimization of memory limit and can automatically optimize the design of a calculation limit, and thus the workload is greatly reduced, and a certain acceleration performance can be obtained. In 2021, Enrico Calore et al. [105] optimized the neural network compiler by pipeline design and used the roof-line model to explore the balance between computational throughput and bandwidth ceiling so that FPGA could perform better.

### 4.3. Memory and Access Optimization

Data storage and access is an essential part of neural networks. A large amount of data will occupy limited memory space. At the same time, a large amount of memory access will greatly increase the execution time of network models and reduce the computational efficiency. In the aspect of memory and access optimization, ping-pong cache, data reuse, standard data access, and so on are usually used for optimization.

### 4.3.1. Ping-Pong Cache

The ping-pong cache is a commonly used data flow control technique, which is used to allocate the input data flow to two random access memory (RAM) buffers in equal time through the input data selection unit and realize the flow transmission of data by switching between the two RAM reads and writes.

As shown in Figure 10, a~c describes the complete operation process of completing the cache of data in RAM A and B and the output of data. Each letter represents the input and output of data at the same time, the inward arrow represents the cache of data, and the outward arrow represents the output data. The specific caching process is as follows: The input data stream is allocated to two data buffers in equal time through the input data selection unit, and the data buffer module is generally RAM. In the first buffer cycle, the input data stream is cached to the data buffer module RAM A. In the second buffer cycle, the input data stream is cached to the data buffer module RAM B by switching the input data selection unit, and the first cycle data cached by RAM A is transmitted to the output data selection unit. In the third buffer cycle, the input data stream is cached to RAM A, and the data cached by RAM B in the second cycle is passed to the output data selection unit through another switch of the input data selection unit.

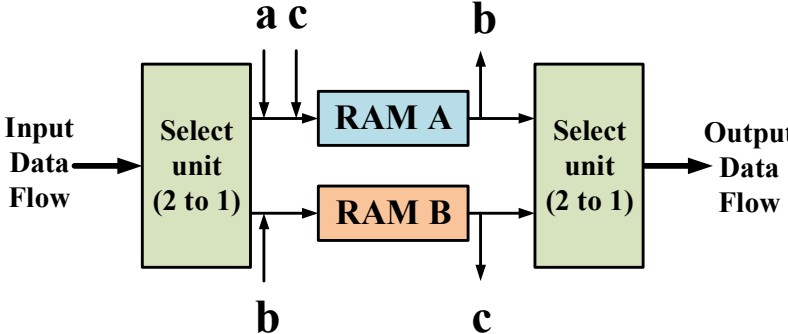

**Figure 10.** Ping-pong cache description.

In this regard, many scholars also put forward their neural network optimization scheme on the basis of this technology. In 2019, Yingxu Feng et al. [106] applied the ping-pong cache to the optimization of the pulse compression algorithm in open computing language (OpenCL)-based FPGA, which plays an important role in a modern radar signal processing system. By using the ping-pong cache of data between the dual cache matched filter and the Inverse fast Fourier transform (IFFT), 2.89 times speedup was achieved on the Arria 10 GX1150 FPGA compared to the traditional method. In 2020, Xinkai Di et al.

[107] deployed GAN on FPGA and optimized it with Winograd transformation and ping-pong cache technology in order to achieve an average performance of 639.2 GOPS on Xilinx ZCU102. In the same year, Xiuli Yu et al. [108] implemented the deployment of the target detection and tracking algorithm on Cyclone IV FPGA of ALTERA Company, completed the filtering and color conversion preprocessing of the collected images, and used the ping-pong cache to solve the frame interleave problem of image data, improve the speed and real-time performance of image processing, and reduce the energy consumption. It met the requirements of miniaturization, integration, real time, and intelligent development of a visual processing system. In 2022, Tian-Yang Li et al. [109] studied the preprocessing of Joint Photographic Experts Group (JPEG) images; optimized the inverse discrete cosine transform (IDCT) algorithm, which is time-consuming in JPEG decoding; and used the ping-pong cache to store the transition matrix. The throughput of 875.67 FPS (frame per second) and energy efficiency of 0.014 J/F were achieved on a Xilinx XCZU7EV.

### 4.3.2. Data Reuse

Data reuse is simply the reuse of data. As can be seen from the roof-line model, when the data reuse rate was low, bandwidth became the bottleneck affecting performance, and the computing resources of FPGA were not fully utilized. Therefore, through data reuse, the actual application bandwidth of memory is greater than the theoretical bandwidth, which increases the upper limit of the bandwidth and also reduces the storage pressure of memory and the amount of data cache and data exchange, so as to reduce the unnecessary time spent.

As shown in Figure 11, in the data operation of neural networks, data reuse is usually divided into input feature map reuse, filter reuse, and convolutional reuse. An input feature map reuse is the reuse of the same input and replacing it with the next input after all the convolution kernels are computed. A filter reuse means that if there is a batch of inputs, the same convolution kernel for the batch is reused and the data are replaced after all the inputs in the batch are calculated. Convolutional reuse uses the natural calculation mode in the convolutional layer and uses the same convolution kernel to calculate the output of different input map positions. In general, these three data reuse modes reuse data at different stages in order to achieve the purpose of increasing performance. Of course, data reuse can also be divided by time reuse and space reuse. For example, if the data in a small buffer is repeatedly used in multiple calculations, it is called time reuse. If the same data are broadcast to multiple PEs for simultaneous calculation, it is called space reuse.

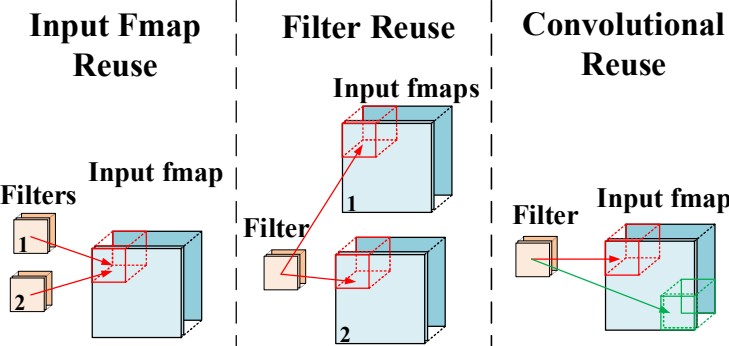

**Figure 11.** Common data reuse modes.

In the FPGA-based deep neural network acceleration experiments, a large part of the acceleration was based on data reuse to increase the bandwidth upper limit and reduce the amount of data cache and data exchange. For example, in 2020, Gianmarco Dinelli et al. [112] deployed the scheduling algorithm and data reuse system MEM-OPT on a Xilinx

XC7Z020 FPGA and evaluated it on LeNet-5, MobileNet, VGG-16, and other CNN networks. The total on-chip memory required to store input feature maps and cumulative output results was reduced by 80% compared to the traditional scheme. In 2021, Hui Zhang et al. [110] proposed an adaptive channel chunking strategy that reduced the size of on-chip memory and access to external memory, achieving better energy efficiency and utilizing multiplexing data and registering configuration in order to improve throughput and enhance accelerator versatility. In 2022, Xuan-Quang Nguyen et al. [111] applied the data reuse technology that were based on time and space to the convolution operation in order to improve data utilization, reduce memory occupancy, reduce latency, and improve computational throughput. The same convolution calculation was about 2.78 times faster than the Intel Core i7-9750H CPU and 15.69 times faster than the ARM Cortex-A53.

### 4.3.3. Standardized Data Access and Storage

A large amount of data operation and frequent data access are the problems that neural networks must encounter when they are deployed on portable systems such as FPGA. In the optimization design of neural networks that are based on FPGA, FPGA is usually used as a coprocessor, that is, the CPU writes the instructions to the memory, and then the FPGA reads and executes the instructions from the memory unit, and following this, writes the calculation results to the memory. Therefore, by standardizing data access, the read and write efficiency of data can be improved, and the actual bandwidth upper limit can be increased.

In the aspect of standardizing data access, scholars often cut the feature map into small data blocks stored in discontinuous addresses for feature mapping to standardize data access patterns. Takaaki Miyajima et al. [113] standardized data access by partitioning storage space and separately accessing sub-modules of each memory space in order to improve effective memory bandwidth. Bingyi Zhang et al. [114] matched the limited memory space on FPGA by data partitioning, eliminated edge connections of high nodes by merging common adjacent nodes, and normalized data access by reordering densely connected neighborhoods effectively, which increased data reuse and improved storage efficiency. Some scholars have used custom data access paths in order to ensure the timely provision of parameters and intermediate data in model inference, as well as to make full use of local memory in order to improve inference efficiency and reduce the amount of external memory access, thus improving bandwidth utilization [115].

## 5. Design of the DNN Accelerator and Acceleration Framework Based on FPGA

In the optimization design of neural networks that are based on FPGA, some FPGA synthesis tools are generally used. The existing synthesis tools (HLS, OpenCL, etc.) that are highly suitable for FPGA greatly reduce the design and deployment time of neural networks, and the hardware-level design (such as RTL, register transfer level) can improve the efficiency and achieve a better acceleration effect. However, with the continuous development of neural networks, its deployment on FPGA has gradually become the focus of researchers. This further accelerates the emergence of more accelerators and acceleration frameworks for neural network deployment on FPGA. This is because with the acceleration of the specific neural network model, the idea is the most direct, and the design purpose is also the clearest. These accelerators are often hardware designs for the comprehensive application of the various acceleration techniques described above. When used in specific situations, such accelerators usually only need to fine-tune the program or parameters to be used, which is very convenient [13]. Figure 12 shows the number trend of relevant papers retrieved by the FPGA-based neural network accelerator on Web of Science by August 2022. The average number of papers published each year is about 140. It can be seen that the research on the FPGA accelerated neural network has attracted increasingly more attention in recent years, which is introduced below.

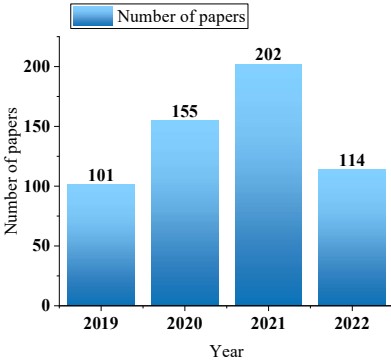

**Figure 12.** The most recent number of papers on the FPGA neural network accelerator in Web of Science.

*5.1. FPGA Accelerator Design for Different Neural Networks*

With the continuous development of deep learning, neural networks have achieved great success in various fields, such as image, speech, and video, and neural networks are developing towards deeper and more complex network design. The way in which to deploy more complex neural networks on FPGA and meet certain speed requirements has become the focus of researchers. In the existing research, a large number of neural network accelerator designs that are based on FPGA have emerged. The main one is the CNN accelerator that is based on FPGA, the RNN accelerator based on FPGA, and the GAN accelerator based on FPGA. The following is a detailed introduction.

5.1.1. The CNN Accelerator Based on FPGA

In the design of many accelerators for FPGA-accelerated convolutional neural networks, most of them focus on improving the network computing power, data transmission speed, and memory occupancy. Scholars usually improve the parallel computing ability of convolutional neural networks in order to improve the computational efficiency and reduce the amount of data and data access in order to solve the problem of large data transmission overhead and large data volume.

In order to solve the problem of heavy computational workload, which limits the deployment of deep convolutional neural networks on embedded devices, the hardware structure shown in Figure 13 was designed in [116]. In this hardware architecture, all buffers use the ping-pong data transfer mechanism to mask the data transfer time with the computation time to improve the performance.

The computing engine (CE) is mainly composed of two computing unit (CU) arrays and several register arrays. Inside each computing unit is a "tree" structure of the bottom multiplier combined with the multilayer adder. Each CU array has 224 CU and can be configured to compute, in parallel, three different parallel modes of output feature maps of the dimension $4 \times 14 \times 4$, $16 \times 14 \times 1$, or $32 \times 7 \times 1$. The flexible configuration ensures high CU utilization in different convolution parameters, so as to maintain a high operation speed. The register array includes the input feature map register (I-REG), the weight register (W-REG), the partial sum register (PS-REG), the output feature map register (O-REG), and the pooling register (PL-REG). On-chip buffers include buffers for input feature map (IBUF), weight (WBUF), partial sum (PBUF), and output feature map (OBUF). Each buffer consists of multiple blocks of RAM, which allow more data to be read or written simultaneously to improve read/write efficiency.

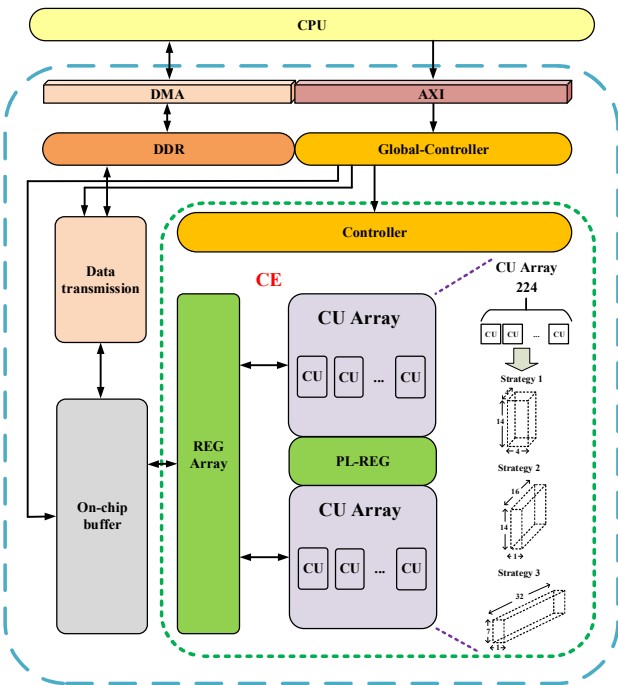

**Figure 13.** Accelerator architecture based on a high-speed computing engine (CE) [116].

In this framework, the image and weight data were first added to the double data rate SDRAM (DDR) by the CPU-controlled direct memory access (DMA), and the top controller of the accelerator was configured, through which the control instructions were assigned to each module. The data transfer module reads the input data and weight parameters from the DDR to the on-chip buffer. Then, the computing engine (CE) performs the convolution operation, and the final result is written back to the DDR through the data transmission module. The architecture successfully deployed VGG-16, ResNet50, and YOLOv2-Tiny lightweight networks on a Xilinx ZynQ-7 ZC706 evaluation version operating at 200 MHz. The performances of 163 GOPS and 0.36 GOPS/DSP were achieved on VGG-16 with only 448 DSP resources. The performances of 0.24 GOPS/DSP and 0.27 GOPS/DSP were achieved on ResNet50 and YOLOv2 Tiny, respectively, achieving a better balance between hardware resource consumption, performance, and reconfigurability compared to previous work.

Although this structure design achieves better performance and less resource consumption, on the whole, this structure does not consider the model parameter optimization of the fully connected layer, and the transmission overhead incurred when processing a large amount of data may limit the actual performance. In addition, this "tree" structure of a computing unit array can only process regular convolution operations. Although it can be configured into three different parallel modes in order to enhance the adaptability of the convolution layer to a certain extent, it is still not suitable for the convolutional neural network after sparse processing. Therefore, with the application of increasingly more irregular convolution operations, the usage scenarios of this hardware structure may be limited.

Some research has focused on the compression of neural network models by using compression techniques such as quantization and pruning operation in order to reduce the amount of data, reduce the memory footprint, and improve the data operation rate. Reference [74] adopted a hardware-friendly quantization scheme suitable for Gaussian-like weight distribution, that is, an intra-layer multi-scheme quantization framework integrated with SP2 and a fixed-point scheme. On Zynq XC7Z020 and XC7Z045, the

performance was improved by 2.1–4.1 times compared with using the DSP alone for all multiplication operations.

References [117–119] all adopted the method of mixed precision quantization, adopting different quantization strategies according to different data accuracy requirements, making the delay lower and the reasoning accuracy higher. In [120], layered fine pruning was used to optimize VGG13BN and ResNet101, which achieved less than 1% precision loss and greatly improved the operation speed when more than 70% parameters and floating-point arithmetic were cut off. Some scholars combined pruning and quantization; used the hybrid pruning method to compress the model; reduced the data bit width to 8 bits through data quantization; and designed the FPGA accelerator to make CNN more flexible, more configurable, and have a higher performance. From the aspect of data occupancy, a compressed storage and computation fusion (CSCF) algorithm was designed in [121] to compress the input data and improve the processing efficiency. Some scholars used a binary neural network to reduce data accuracy in order to improve data transmission speed, so as to improve the computational efficiency of convolutional neural networks [122].

In addition, the idea of reducing the multiplication operation or logical shift and adder can be used to replace the idea of a multiplier, so as to reduce the occupation of resources and improve the speed of a multiplication operation [74,123]. As shown in Figure 14, the method of reducing the multiplication operation was proposed in [123]. In this example, a traditional 3 × 3 convolution operation will use nine multiplication operations and eight addition operations. The simplified convolution operation reduces the number of multiplication operations by counting the number of numbers appearing for the first time in the convolution kernel. Under the condition that the number of additional operations is unchanged, the number of multiplication operations is reduced to 2. However, in general, the values in the convolution kernel are essentially different, and there are not many repeated terms, as in the example. Therefore, this method may not have a good effect in practical application, unless the repeated terms in the convolution kernel are increased by some means. In [124], the idea of skipping multiplication operation was directly adopted in order to avoid unnecessary computation by skipping multiply accumulate (MAC) with zero weight, so as to improve the operation speed. Of course, the on-chip resources of FPGA can also be divided into multiple small processors in order to improve the computing power [125], or a single neural network model can be distributed to the FPGA cluster in order to improve the overall computing throughput [61].

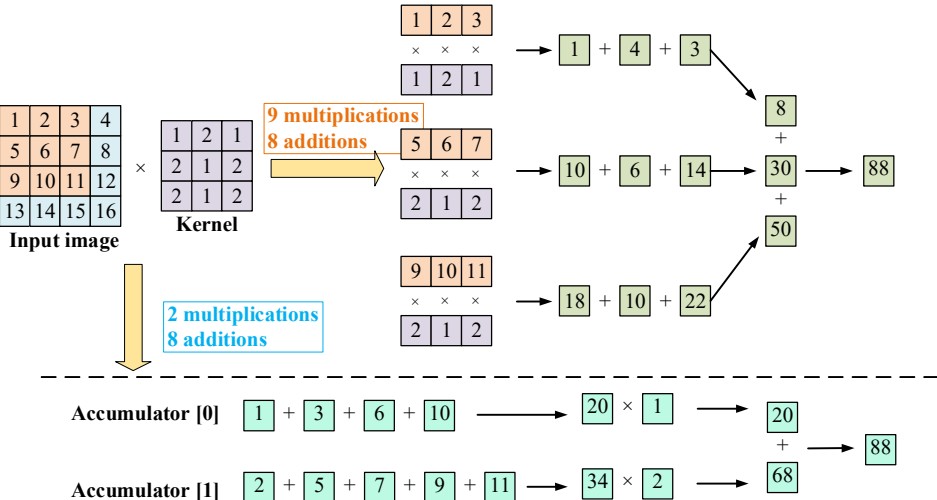

**Figure 14.** A simple operation used to reduce multiplication.

### 5.1.2. The RNN Accelerator Based on FPGA

Because the traditional RNN has the problem of gradient disappearance and explosion, the network performance is not good in the application, requiring long-term input information. Some researchers proposed a variant of RNN, long short-term memory (LSTM), to solve this problem. Although LSTM can solve the gradient disappearance problem, it cannot avoid the gradient explosion problem, and because it introduces many gating units, it leads to the problem of a large number of parameters [126]. Others have proposed the gate recurrent unit (GRU), which, like LSTM, has been put forward to solve problems such as long-term memory and gradients in back propagation [127]. In many cases, the GRU and LSTM are practically similar in performance. The largest difference between GRU and LSTM is that GRU combines the forgetting gate and the input gate into one "update gate," and the network does not provide additional memory states. Instead, the output results are continuously recycled back as memory states, and the input and output of the network become particularly simple. However, GRU still cannot completely solve the problem of vanishing gradient. At present, the literature in this field mainly focuses on LSTM and GRU models. The design of an FPGA-based recurrent neural network accelerator is essentially the same as that of a convolutional neural network accelerator, which are carried out from the perspective of improving the computing performance and resource utilization of FPGA and reducing the storage pressure.

In order to reduce RNN memory overhead for acoustic task, the [101] used a quantitative method to reduce the activation volume, wherein the time-consuming floating-point arithmetic was replaced by the faster floating-point arithmetic, greatly improving the network operation speed and joining the parameters of the mechanism of sharing and pipelined design in order to increase parallelism and reduce storage, further improving the network throughput and reducing the delay. However, the data quantized by this scheme is still the floating point. Fixed-point quantization can greatly improve the speed of data operation and reduce the memory overhead without much impact on the accuracy. Due to the feedback dependence of LSTM, high parallelism cannot be achieved on general processors such as the CPU and GPU. Reference [128] proposes an implementation scheme of the LSTM network acceleration engine based on FPGA by taking advantage of FPGA's characteristics of low power consumption, low delay, and good parallelism. The optimization is carried out by a fixed-point algorithm, a pulsating array, and a nonlinear function lookup table.

Compared with CPU and GPU implementation, the FPGA has the lowest power consumption, the lowest delay, and the highest energy efficiency ratio. This scheme is shown in Figure 15.

The accelerator architecture firstly divides the input matrix and weight matrix into small blocks in advance, increases the parallelism of the matrix operation, and combines the pulsating array algorithm in order to accelerate the computation. Moreover, the problem of low data reading efficiency caused by data discontinuity during sequential data reading is solved by rearranging the elements in the matrix. Among them, the traditional processing element (PE) from memory read data performs various calculations and then writes the results back to the storage architecture, wherein the pulsating array appears in the form of lines, and each PE calculation no longer relies on memory access, only the first array PE in terms of reading data from memory, after processing the results directly to the next PE. At the same time, the first PE can read the next data from memory, and so on, until the last PE in the array writes the result back to memory every clock cycle. In this structure, each PE is processed in parallel, which greatly reduces the number of memory accesses. Secondly, in view of the difficulty of realizing activation functions such as Tanh/Sigmoid in FPGA, a look-up table (LUT) and polynomial approximation are used instead. Finally, a fixed-point operation is carried out on the data in the unit computation, and a fixed-point number is used instead of a floating-point number for multiplication and addition operation, which greatly saves the resources of FPGA, improves the computation speed, and reduces the delay.

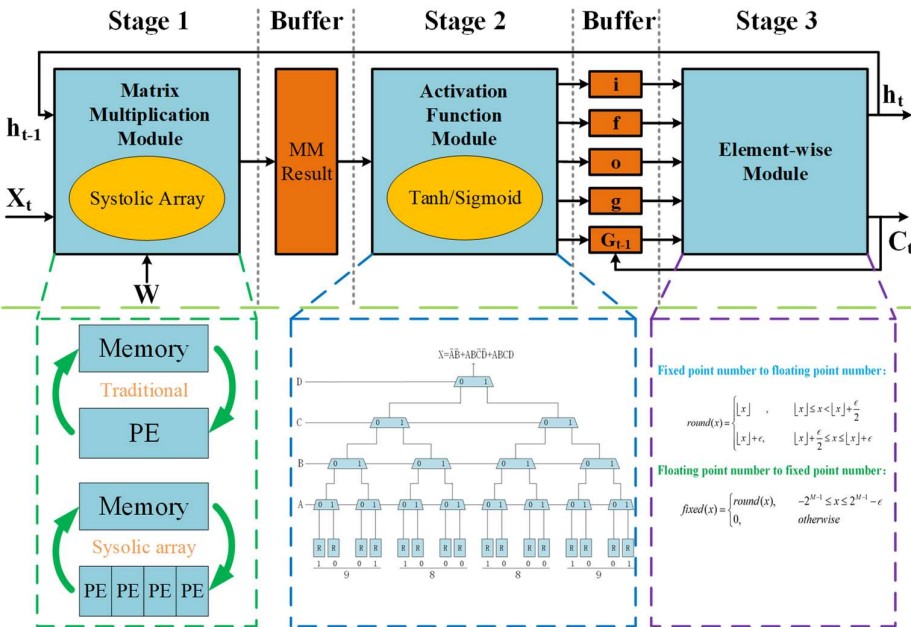

**Figure 15.** System structure of the LSTM accelerator.

There are also some studies devoted to the compression of neural networks. Reference [129] proposed a hybrid iterative compression algorithm (HIC) for LSTM/GRU to compress LSTM/GRU networks. The utilization of block RAM (BRAM) is improved by rearranging the weights of the data stream of the matrix operation unit based on block structure matrix (MOU-S) and by fine-grained parallel configuration of matrix vector multiplication (MVM). At the same time, the combination of quantization and pruning greatly reduces the storage pressure. In order to accelerate the inference of LSTM, the authors of [130] produced an improvement on the basis of the previous compression method for weight pruning, called "bank balanced sparsity (BBS)." In view of the problem that BBS requires a large amount of extra memory overhead to store indexes, the compression efficiency was greatly limited. The implementation of a shared index bank balanced sparsity compression method (SIBBS) reduces the memory overhead by 2–8 times and achieves up to 79.5 times delay reduction with almost the same accuracy.

### 5.1.3. The GAN Accelerator Based on FPGA

GANs mainly consist of a generator and a discriminator, producing better output through mutual game learning between generator G and discriminator D. In the original GAN theory, both G and D are not required to be neural networks, as long as the corresponding functions that are generated and discriminated against can be fitted. However, in practical application, deep neural networks used as G and D generally, so in the GAN accelerator design based on FPGA, it is basically aimed at the generator and the discriminator to optimize. Therefore, it is essentially optimized for CNN, RNN, or other neural network, mainly from the lower network number, wherein the accelerators are designed from the perspectives of reducing storage pressure, reducing computational complexity, and improving network computing speed.

Reference [69] focused on optimizing multiplicative and cumulative operation in GANs to reduce power consumption. In some works, the binarized GAN [131] and the ternary GAN [132] were used to reduce the precision of weight data, so as to improve the operation speed. In [133], the deconvolution operation in GAN was optimized to achieve higher throughput and resource utilization. On the basis of optimizing deconvolution operation, the authors of [134] used the pruning operation to reduce the computational complexity and the scale of the network model. Some scholars used a new fast transformation

algorithm (FTA) for deconvolution calculation, which solved the computational imbalance problem well and eliminated the extra memory requirement of overlapping parts and sums [135]. In addition, the authors of [136] optimized the intensive convolution computation in GAN and the invalid computation caused by the insertion of a "zero" value in the deconvolution operation, using the reconfigurable mechanism to switch the two convolution functions flexibly on the same process element group (PEG), thus increasing the utilization of on-chip resources and improving parallelism. In order to solve the problem of high computational complexity and the need to store a large amount of intermediate data in the training of GAN on an embedded platform, a reconfigurable accelerator based on FPGA was also proposed in [137] for effective GAN training.

The acceleration framework is shown in Figure 16. Firstly, the cascade fast FIR (finite impulse response) algorithm (CFFA) is optimized for GAN training, and the fast convolution processing element (FCPE) based on the optimization algorithm is introduced to support various computing modes during GAN training. In the input prefetcher module and weight prefetcher module, 16-bit fixed point and 8-bit fixed point were used, respectively, to process data, greatly improving the data processing speed. Finally, the architecture achieved 315.18 GOPS performance and 83.87 GOPS/W energy efficiency on the Xilinx VCU108 FPGA platform with 200 MHz operating frequency. Experiments showed that the architecture was able to achieve high energy efficiency with less FPGA resources. Although the architecture can effectively avoid the large consumption of hardware resources by cascading small parallel FIR structures to larger parallel FIR structures, this operation greatly increases the delay. When the number of cascades reaches a certain limit, the performance loss caused by the delay is incalculable.

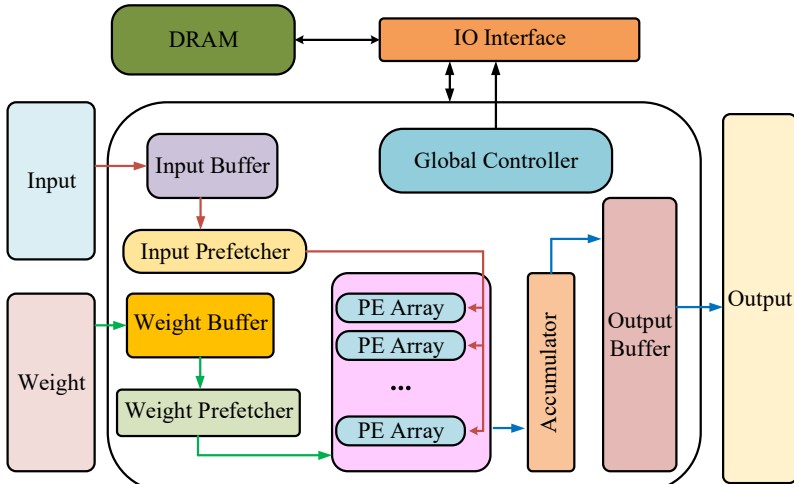

**Figure 16.** The GAN acceleration framework.

Table 2 shows the performance comparison of different neural networks deployed with different optimization methods on different FPGA platforms in recent years. From Table 2, we can see that the use of Winograd and other convolution optimization technologies can bring about great performance gains because the most important operation in the neural network is the convolution operation, which occupies most of the operation of the whole neural network and consumes the largest number of resources and has the longest processing time. Therefore, the optimization for a convolution operation can bring about much higher performance gains than other optimization techniques.

**Table 2.** Performance comparison of different networks deployed using different optimization methods on different FPGA platforms in recent years.

| Reference | Platform | Freq (MHz) | Neural Network | Methodology | Performance |
|---|---|---|---|---|---|
| [107] | ZCU102 FPGA | 200 | GAN | Winograd, Pipeline, Fixed-16 | 639.2 GOPS |
| | VC706 FPGA | 167 | | | 162.5 GOPS |
| [138] | XCKU9P FPGA | 200 | LSTM | Pruning, Fixed-16 | 200 GOPS |
| [74] | XC7Z020 FPGA | 100 | ResNet-18 | Mixed quantization | 77.0 GOPS |
| | | | MobileNet-v2 | | 71.8 GOPS |
| | XC7Z045 FPGA | | ResNet-18 | | 359.2 GOPS |
| | | | MobileNet-v2 | | 326.9 GOPS |
| [139] | Stratix10 GX2800 FPGA | 260 | LSTM | Optimize multiplication, Fixed-8 | 7041 GOPS |
| [131] | Xilinx ADM-PCIE-7V3 FPGA | 200 | Vanilla LSTM | Pruning, optimize multiplication | 379507 FPS |
| | | | GRU | | 378788 FPS |
| [117] | XC7Z020 FPGA | 100 | ResNet-18 | Mixed quantization | 101.3 GOPS |
| | | | MobileNet-V2 | | 80.1 GOPS |
| | XC7Z045 FPGA | | ResNet-18 | | 446.8 GOPS |
| | | | MobileNet-V2 | | 363.5 GOPS |
| [140] | ZC706 FPGA | 200 | VGG-16 | Pipeline | 706 GOPS |
| | | | AlexNet | | 624 GOPS |
| | | | ZFNet | | 648 GOPS |
| | | | YOLO | | 702 GOPS |
| [135] | Stratix 10SX FPGA | 185 | GAN | Convolution optimization, memory optimization | 2211 GOPS |
| [130] | XCKU115 FPGA | 200 | LSTM | Pruning, Fixed-8, Fixed-12 | 1424.8 GOPS |
| [141] | Arria 10 GX1150 FPGA | 220 | Bayes-VGG11 | Optimize calculation | 533.75 GOPS |
| | | | Bayes-ResNet18 | | 1590 GOPS |
| | | | Bayes-C3D | | 1449 GOPS |
| [64] | Xeon 4116 CPU + Arria10 GX1150 FPGA | 163 | VGG + BiLSTM | Winograd, Fixed-8, Fixed-16 | 1223.53 GOP/s |
| [142] | XCZU7EV FPGA | 640 | MNIST LSTM | Memory optimization, 16–27 fixed | 44.5 GOPS |
| | | 420 | Character LSTM | | 363.7 GOPS |

*5.2. Accelerator Design for Specific Application Scenarios*

In the practical application of neural networks, people often customize FPGA accelerators with required functions according to specific application scenarios. Since accelerators customized according to specific application scenarios are relatively easy to design and can effectively solve the corresponding problems, this method is commonly used to accelerate neural networks in specific application scenarios, especially in speech recognition, image processing, natural language processing, and other fields.

### 5.2.1. FPGA Accelerator for Speech Recognition

Speech recognition is a technology that enables machines to automatically recognize and understand human spoken language through speech signal processing and pattern recognition. In short, it is the technology that allows a machine to transform a speech signal into a corresponding text or command through the process of recognition and understanding. Speech recognition has been applied in many fields, including speech recognition translation, voice paging and answering platforms, independent advertising platforms, and intelligent customer service. Speech recognition has strong real-time performance and high delay requirements, and thus it generally relies on the FPGA platform.

In terms of accelerating the application of speech recognition, the authors of [143] focused on the preprocessing stage of speech signals and proposed the use of a GAN to enhance speech signals and reduce noise in speech signals, so as to improve speech quality and facilitate speech recognition and processing. In order to reduce energy consumption and improve the speed of speech recognition, the authors of [138] proposed an FPGA accelerator structure called balanced row dual-ratio sparsity inducing pruning algorithm (BRDS) for speech recognition, as shown in Figure 17. The accelerator compresses the LSTM network by pruning algorithm in order to reduce computational complexity and uses data reuse and pipeline design to achieve low power consumption and low delay. However, the acceleration architecture does not consider the problem of multi-core parallel load imbalance after LSTM model compression, which may occur in the actual use, thus affecting the performance of the whole accelerator.

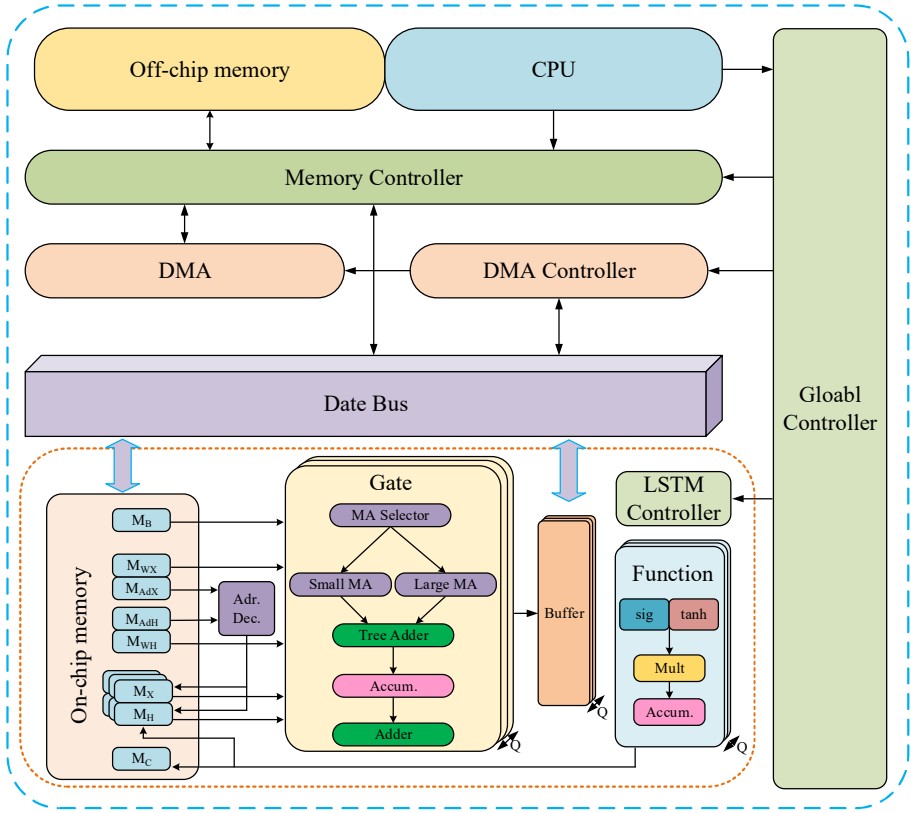

**Figure 17.** BRDS: FPGA accelerator for speech recognition.

To solve this problem, as early as 2017, the ESE system architecture proposed in [58] can be well solved. The proposed hardware structure is shown in Figure 18. The architecture takes the ESE accelerator on FPGA as the core and solves the problem of multi-core parallel load imbalance by using multiple channel units composed of processing units and

activation vector queue units. The processing unit adopts a double-buffer structure and ping-pong data transmission mode to improve the execution efficiency. The faster processing unit can obtain data from the activation vector queue unit and continue to work without waiting for other processing units. At the same time, the problem of workload imbalance between different processing units is solved. Compared with CPU (i7 5930 K) and GPU (GTX Titan X), the ESE architecture on the Xilinx XCKU060 FPGA platform was 43 times and 3 times faster than CPU and GPU platforms, respectively, and the energy efficiency ratio was 40 times and 11.5 times higher, respectively. However, the architecture design is relatively complex, involving an FPGA hardware accelerator, a CPU software program, and external memory modules. Task allocation and scheduling coordination among the three modules have become the bottleneck restricting the performance of the architecture [13].

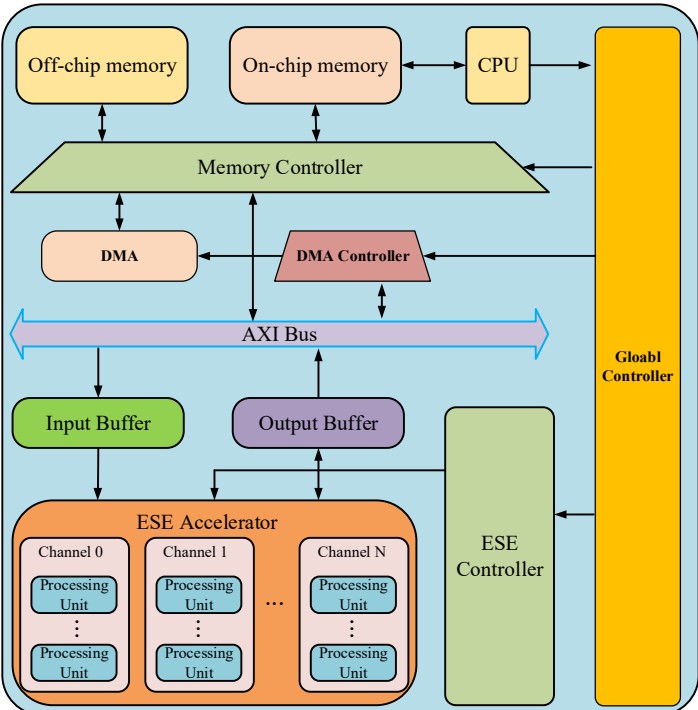

**Figure 18.** ESE system architecture.

### 5.2.2. FPGA Accelerator for Speech Recognition

Image processing refers to the process of extracting, analyzing, and processing image information by computer, which is one of the earliest application fields of deep learning. Image processing techniques include point processing, group processing, geometric processing, and frame processing. The main research contents include image enhancement, image restoration, image recognition, image coding, image segmentation, among others. Since 2012, ImageNet competition, image recognition, and processing technology have been widely used in image classification, face recognition, and other fields. In the design of an FPGA accelerator for image processing, the accelerator is designed mainly for the purpose of reducing the amount of data, memory, and computation demand [97,144–146].

Some scholars adopted the idea of increasing intra-layer and inter-layer parallelism to speed up computation and reduce memory consumption [147]. Some scholars used pipeline technology to deploy VGG 16 on an Intel Arria 10 FPGA and achieved a throughput of 736.9 GOPS. Reference [148] focused on optimizing the multiplication operation in the convolution layer and adopted the deep separable convolution (DSC) layer to greatly reduce the network complexity while maintaining the classification accuracy. The application of MobileNetv2 on FPGA achieved a throughput of 413.2 GOPS. Different from the

general direction of optimizing convolution computation, the authors of [109] focused on optimizing non-convolution operations in order to improve throughput and energy efficiency. Some scholars based their work on quantization technology in order to reduce the amount of neural network data, so as to accelerate the calculation of face direction recognition [149]. Referenced [150] designed FPGA accelerators from the perspective of reusing hardware resources, so as to reduce the utilization of FPGA resources.

Reference [151] quantitatively analyzed the influence of aggregation and combination order of a change graph neural network (GNN) on performance, finding that the optimal execution order was not fixed. In addition, there were some problems, such as memory bottleneck, insufficient parallel computing capacity in combination stage, load imbalance, and the influence of feature dimension change between layers on graph partitioning efficiency. In order to solve these problems, an adaptive GNN accelerator framework (AGA) was proposed, which is able to provide flexible workflow support. Different commands are executed in different layers, memory subsystem and sparse elimination technology are used to alleviate memory bottlenecks, more parallel design is added to improve the utilization of computing resources, and the inter-layer graph partitioning strategy is optimized. At the same time, hardware resources are dynamically allocated in order to achieve load balance.

As shown in Figure 19, AGA consists of off-chip memory, a memory controller, DMA, a DMA controller, on-chip buffer, a processing module (PM), and a workflow controller. Although compared with CPU and GPU, this architecture had 665 times and 24.9 times the performance speedup ratio, respectively, and 3180 times and 138 times the energy efficiency speedup ratio, respectively, this architecture only performs sparse data processing without optimizing the most resource-consuming and time-consuming multiplication operations, which greatly limits the performance of the accelerator.

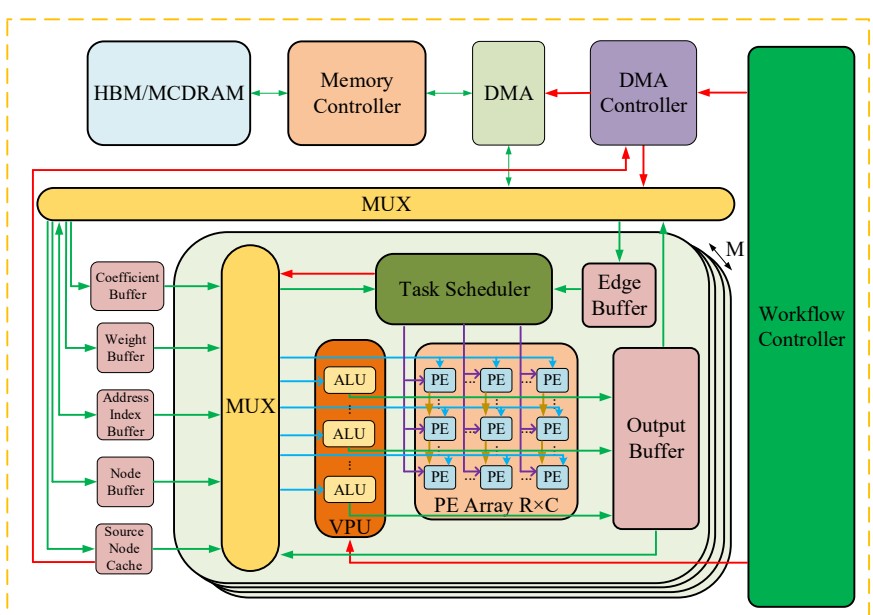

**Figure 19.** The Adaptive GNN accelerator (AGA) framework.

5.2.3. FPGA Accelerator for Natural Language Processing

Natural language processing (NLP) is a technology that uses the natural language used by humans to communicate with machines. It studies various theories and methods that can realize effective communication between humans and computers using natural language, and it is also one of the important application fields of deep learning.

In terms of accelerating NLP, the authors of [152] point out that although the language representation based on transformer [153] has achieved the most advanced

accuracy on various NLP tasks, the deployment of large model networks has always been a challenge for resource-constrained computing platforms. Weight pruning is used to reduce weight parameters, which can effectively compress network models and improve network performance. In order to solve the problem of large storage demand and network performance degradation caused by a large number of parameters when RNN is applied to natural language processing, the authors of [154] focused on the computing resource demand of RNN and adopted fixed-point quantization technology in order to design an FPGA accelerator, which reduced the memory consumption by 90%, and the accuracy loss was less than 1%.

Different from [154] on quantifying input data, some scholars have devoted themselves to NLP task optimization based on the BERT (bidirectional encoder representation from transformers) network model [155] and have adopted the idea of full quantization to a design accelerator. Not only input data but also weights, activations, Softmax, layer normalization, and all the intermediate results are quantified in order to compress the network and improve performance [156].

Some scholars put forward an optimization scheme for heterogeneous NLP models that can automatically improve the performance of NLP models on CPU, GPU, and FPGA by identifying key performance operations and their performance problems in various NLP models [157]. In order to accelerate the inference of the BERT model, the authors of [158] proposed an FPGA-based overlay processor T-OPU for the inference of the BERT model in view of the large number of natural language models and the update of fast characteristics. T-OPU is composed of a data extraction unit, a matrix multiplication unit (MMU), an output control unit, a nonlinear vector memory (NVM) unit, and a nonlinear vector (NV) unit, which can efficiently execute various natural language processing models.

### 5.3. FPGA Accelerator for Optimization Strategy

In the neural network accelerator design based on FPGA, one often needs to study the characteristics of the neural network and carry out targeted optimization on the basis of the characteristics of its operation, wherein the reasonable optimization strategy can significantly improve the performance of the accelerator and resource utilization, the common optimization strategy for the calculation of the optimization, and optimization for storage, among other improvements.

#### 5.3.1. Optimization for Calculation

In neural networks, the convolution layer and convolution operation are usually the focus of our optimization, especially in convolutional neural networks applied to image processing. The convolutional operation is more intensive, the processing time is longer, and the resources are more occupied, which limits the deployment of convolutional neural networks in edge platforms such as FPGA.

The usual optimization direction is to increase the parallelism between different layers of the neural network model, the parallelism between different output feature maps, the parallelism between different pixels, and the parallelism between different pixels in order to shorten the processing time and improve the network performance. It can also use circular flow and loop unfolding to build a deeper pipeline to shorten the overall execution time of the network and reduce the time overhead. The matrix cycle is divided into smaller modules by cyclic block technology, and each module can be computed in parallel to improve computing parallelism. Moreover, through data reuse, the number of memory accesses can be reduced, so as to improve the computational efficiency. Reference [159] used data reuse and task mapping technology in order to improve memory bandwidth and throughput, so as to improve design efficiency. In [160], the parallelism between channels was utilized through cyclic unrolling, so as to improve the network running speed and network performance. On the FPGA operating frequency of 110 MHz, the peak performance of 213.7 GOP/s and top-5 accuracy of 79.05% were achieved, and the energy efficiency ratio was at least 4.3 times that of other CPU and GPU platforms.

### 5.3.2. Optimization for Storage

Due to the large amount of data and parameters in neural networks, problems of parameter and data storage are inevitable when neural networks are deployed. In the optimization of storage, the strategy of reducing data precision is usually used to reduce the amount of data, so as to improve the operation speed. The data reuse method can be adopted in order to greatly reduce the data storage and the memory space occupation. For example, ShortcutFusion, an accelerator optimization tool proposed in [161], reduces memory footprint by maximizing on-chip data reuse under given resource constraints. The data processing time was found to be 2.8 times faster than NVIDIA RTX 2080 Ti, and the energy efficiency ratio was nearly 10 times higher on a Xilinx KCU1500 FPGA. Some scholars have used a memory management framework to realize automatic data layout and to solve the memory competition problem of the CPU-FPGA heterogeneous system by analyzing cross-layer memory competition. By automatically generating the optimal data placement strategy and the cache partition mechanism of the parallel execution kernel, the performance of both the FPGA kernel and the CPU kernel in the heterogeneous system is improved [115].

### 5.4. Other FPGA Accelerator Designs

In addition to the above accelerator design, there are some other accelerator designs, such as the accelerator design based on the hardware template. By using a ready-made hardware template to design the accelerator, it is only necessary to improve the module and configure parameters according to specific problems, which greatly improves the deployment of the model [162].

Some scholars design accelerators for different algorithms and optimize them according to the characteristics of each algorithm. For example, by analyzing the parallelism of the SAR (synthetic aperture radar systems) real-time back projection algorithm, a fully parallel processing architecture of the back projection (BP) algorithm based on FPGA is proposed in [163]. In the BP imaging algorithm, fixed-point quantization and distributed memory technology are combined. Compared with other algorithms implemented on FPGA, DSP, GPU, and CPU without optimization, the optimized algorithm on the Xilinx XC7VX690T FPGA has more obvious characteristics of output images and a more obvious focusing effect. There was also a study [164] that used OpenCL language to optimize the FPGA-based fuzzy C-means (FCM) algorithm in order to increase the parallelism of the model by enabling advanced compilers or synthesis tools to operate task parallel models and create efficient designs, thus improving the algorithm speed. Compared with the conventional single-core CPU, the optimized FCM algorithm can achieve 89 GFLOPS, which is 186 times higher than the CPU.

### 6. Current Challenges and Future Outlook

Deep learning takes neural networks as the main model. With the continuous development of deep learning, the research on accelerating various neural networks has attracted increasingly more attention. Although FPGA has made some achievements in accelerating various neural networks with its advantages such as being reconfigurable, having low energy consumption, and having low delay, it also has some shortcomings such as difficult hardware programming, a long reconstruction time, and a high learning cost. Therefore, FPGA needs a certain amount of time and technical support to achieve a wider application range, be used in higher application scenarios, and become more well-known to people. According to the current development status, in terms of the FPGA-accelerated neural network, the key research directions for the future are mainly as follows:

(1)  Improve the FPGA ecological environment. At present, the ecosystem of FPGA is relatively closed, and a few companies such as Xilinx and Intel control the main industrial chain set for FPGA. The lack of industry standards and basic specifications produces FPGA learning and development problems, such as high threshold, long

cycle, and limited use of development tools emerging one after another. The wide application of deep neural networks in various fields has prompted researchers to put forward some performance requirements for deep neural networks. The way in which to use FPGA in order to accelerate deep neural networks and improve the performance of deep neural networks has gradually become the focus of scholars. However, due to the limitations of the FPGA ecosystem, the speed of its application cannot keep up with the development speed of software algorithms, and this also limits the further application of deep neural networks. This is an urgent problem for researchers to solve.

(2)   Train FPGA professionals. At present, there is the complexity of the FPGA learning threshold being high, the content and learning cycle being long, and the existence of problems such as an incomplete training system causing the FPGA to lack professional talent reserves and a shortage of reserve forces, and so on and so forth. With the development of large data and artificial intelligence technology, the demand for professional talents in the FPGA area is becoming increasingly larger, with the problem is becoming increasingly more prominent.

(3)   Optimize the activation function. At present, in the computational optimization of FPGA-based deep neural networks, most of the optimization schemes are for the convolution operation and the cycle part of matrix operation, while there are a few improvement schemes for the activation function optimization, and therefore this is a potential performance breakthrough.

(4)   Optimize convolution and other operations. From this article, reviewing different FPGA platform application optimization technology deployments found in the neural network performance comparison table, we found that use of convolution optimization or other optimal performance gains come about when other forms of computing is the highest, meaning that one can explore the method of updating in order to optimize convolution or other operations, with this gaining better performance.

(5)   Data optimization. By replacing the high-width data with the low-width data, the data processing time can be reduced, and the data storage pressure can be alleviated, so as to improve the acceleration performance of the model, and at the same time, it brings about the disadvantage of precision loss. Later, the research on dynamic precision data quantization can be strengthened in order to make the fixed-point number corresponding to different layers of the neural network have different integer and decimal places, so as to achieve a shorter fixed-point number while maintaining the required higher accuracy and smaller accuracy loss.

(6)   Neural network model optimization. It is also a feasible scheme to improve the performance of neural networks by optimizing the structure of the neural network model or realizing its rapid deployment on the FPGA platform.

(7)   The cluster for FPGA. Through multiple FPGAs speeding up the neural network reasoning, one can achieve higher performance and lower latency, and the difficulties of this method are how to coordinate between each FPGA processing scheduling problem and task assignment problem. The future can be from a variety of fine-grained classification and distribution of weights between the FPGA for optimization, so as to improve the utilization rate of the on-chip memory and reduce storage requirements.

(8)   Multiple FPGA accelerators. Similar to FPGA clustering, acceleration can be achieved by distributing tasks among multiple FPGA accelerators. The difficulty of this method is task scheduling and assignment among accelerators. In the future, we can focus on the effective task allocation, scheduling, and processing methods among multiple FPGA accelerators, so that the acceleration scheme of multiple FPGA accelerators can achieve better acceleration effect.

(9)   Lightweight network. The lightweight network is very suitable for deployment on edge platforms such as FPGA. On the one hand, the deployment difficulty of the lightweight network is greatly reduced, and on the other hand, the lightweight

network is more suitable for edge platforms with fewer available resources, relatively low performance, and high-power consumption requirements. This means the deployment of the lightweight network on FPGA to complete the specified task or replace the network with a lightweight network when the task indicators can be completed, which will be the direction of future research.

As a revolutionary realization method of machine learning, deep learning technology with neural network as the core has broad development prospects in the future. Similarly, various deep neural networks will also face various application scenarios, which have certain requirements on the performance of neural networks and their deployment platforms. At present, on the basis of the advantages of FPGA reconfiguration, low latency, and high parallelism, use of FPGA to accelerate various neural networks has gradually become the choice of most researchers. Although the FPGA ecosystem is still not perfect and there are still many problems to be solved, it can be predicted that, with the passage of time, FPGA-based deep neural network acceleration technology will gradually mature and eventually promote the reform and development of the whole field of artificial intelligence.

### 7. Summary

This paper firstly introduces the development process and application field of some representative neural networks, analyzes and summarizes the development process of neural networks, and divides the neural network process into five stages. It points out the importance of studying deep learning technology based on neural networks, as well as the reasons and advantages of using FPGA to accelerate deep learning. Several common neural network models are introduced. This paper summarizes the current mainstream FPGA-based neural network acceleration technology, method, accelerator, and acceleration framework design and the latest research status, as well as analyzing the performance of various technologies. The techniques with higher performance gain are summarized, and the reasons for this phenomenon are provided. At the same time, the current difficulties in the application of neural networks based on FPGA are pointed out, and the future research directions are prospected. The aim is to provide research ideas for the researchers engaged in the field of neural network acceleration based on FPGA.

**Author Contributions:** Conceptualization, Z.L.; methodology; investigation, C.W.; writing—original draft preparation, C.W.; writing—review and editing, Z.L.; supervision, Z.L.; project administration, Z.L.; funding acquisition, Z.L. All authors have read and agreed to the published version of the manuscript.

**Funding:** This work was supported in part by the National Natural Science Foundation of China under grant 61801319, in part by Sichuan Science and Technology Program under grants 2020JDJQ0061 and 2021YFG0099, in part by Innovation Fund of Chinese Universities under grant 2020HYA04001, in part by the Sichuan University of Science and Engineering Talent Introduction Project under grant 2020RC33, and in part by the Postgraduate Innovation Fund Project of Sichuan University of Science and Engineering under grant Y2022124.

**Conflicts of Interest:** The authors declare that they have no conflict of interest.

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
