# Peer review of "A Review of the Optimal Design of Neural Networks Based on FPGA"

_applsci, doi:10.3390/app122110771_

Round 1

Reviewer 1 Report

The manuscript is well written and well structured. The authors have presented in detail the state-of-the-art techniques for optimal design of neural networks based on FPGA. Some minor changes that can be incorporated are as follows:

Fig. 2, Fig. 4 and Fig. 5 can be replaced with better versions.

The blocks in Fig. 6 should include references in addition to the texts.

Overall a good manuscript with rich reference.

Reviewer 2 Report

This article is organized and written well. Please read and check on the typos in the text, for example, it is "Generative Adversarial Networks (GAN)" not "Generative networks Adversarial Networks (GAN)" on Page 5, Lines 175 and 176. 

Any abbreviations must be defined first before using.

Appropriate citations need to be added to the caption of each figure if it is adopted or modified from the published articles such as Figs. 3, 7, 8, 9, 13, 15 ~ 20. 
